# An improved grasshopper-based MPPT approach to reduce tracking time and startup oscillations in photovoltaic system under partial shading conditions

**Muhammad Shahid Wasim**[1⊙]*, **Muhammad Amjad**[1⊙], **Muhammad Abbas Abbasi**[1‡], **Abdul Rauf Bhatti**[2‡], **Akhtar Rasool**[3‡]*

1 Faculty of Engineering, The Islamia University of Bahawalpur, Bahawalpur, Punjab, Pakistan, 2 Department of Electrical Engineering and Technology, Government College University, Faisalabad, Punjab, Pakistan, 3 Department of Electrical Engineering, University of Botswana, Gaborone UB, Botswana

⊙ These authors contributed equally to this work.
‡ MAA, ARB and AR also contributed to this work.
* shahid.wasim@iub.edu.pk (MSW); rasoola@ub.ac.bw, akhtar@sabanciuniv.edu (AR)

**Data Availability Statement:** All relevant data are within the manuscript.

## Abstract

Global maximum power point (GMPP) tracking under shading conditions with low tracking time and reduced startup oscillations is one of the challenging tasks in photovoltaic (PV) systems. To cope with this challenge, an improved grasshopper optimization algorithm (IGOA) is proposed in this work to track the GMPP under partial shading conditions (PSC). The performance of the proposed approach is compared with well-known swarm intelligence techniques (SITs) such as gray wolf optimization (GWO), cuckoo search algorithm (CSA), salp swarm algorithm (SSA), improved SSA based on PSO (ISSAPSO), and GOA in terms of tracking time, settling time, failure rate, and startup oscillations. For a fair comparison, the PV system is analysed under uniform irradiance and three PSCs having four to six peaks in the power-voltage characteristic curves and using three to six search agents for each SIT. For this purpose, a PV system containing six solar panels has been built using MATLAB/SIMULINK software, and statistical analysis is performed in detail. The results show that the IGOA tracks the GMPP in 0.07 s and settles the output in 0.12 s which is 25% to 96% faster than its counterparts. Moreover, IGOA proves its consistency with a minimal tracking failure rate of 0% for four to six search agents with negligible startup oscillations. This work is expected to be helpful to PV system installers in obtaining maximum benefits from the installed system.

## 1 Introduction

The major share of electrical energy is produced by fossil fuels, which emit greenhouse gases and harm the environment [1]. Harnessing energy from renewable energy (RE) resources has acquired much scientific attention to lessen the negative effects of fossil fuels on the environment. Photovoltaic energy is gaining popularity over other sources of RE such as geothermal,

**Funding:** The author(s) received no specific funding for this work.

**Competing interests:** The authors have declared that no competing interests exist.

wind, biomass, fuel cells, and hydropower due to its universal availability, low emissions, and low operating cost [2]. Partial shading caused by clouds and obscuring objects such as buildings or trees is a common problem in PV systems [3]. Shaded PV modules operate in reverse bias and act as a load that decreases the performance of the system [4]. Although bypass diodes are used to reduce this effect, this causes the characteristic curves to become non-uniform, resulting in the formation of multiple local peaks (LP) [5]. Therefore, it is necessary to track a global peak surrounded by several LPs to improve the performance of the PV system [6].

Numerous methods have been proposed to track the maximum power point (MPP) in the literature, such as offline, online, artificial intelligence (AI) and swarm intelligence techniques [7–11]. Offline approaches are not ideal for high-performance tracking, as we have to separate the load from the PV array to measure the values of open-circuit voltage and short-circuit current [12]. Furthermore, due to the imprecise estimation of the proportionality constant, the operational point is not exactly on the MPP [13]. Online/traditional algorithms are unable to track the global MPP under rapidly changing irradiance and partial shading conditions (PSCs) [14, 15]. The algorithms do not converge on MPP if the irradiance changes within two sample steps [16]. AI methods require hundreds of data sets to train the PV system which is time-consuming and needs much knowledge [17, 18]. The high memory space to save the customized parameters is another disadvantage [19]. Hybrid AI algorithms have received less attention due to their higher costs, computational time, and algorithm complexity [20].

Swarm intelligence techniques (SITs) such as particle swarm optimization (PSO) [21, 22], cuckoo search algorithm (CSA) [23, 24], whale optimization algorithm (WOA) [18], moth flame optimization (MFO) [25], bat algorithm (BA) [26], gray wolf optimization (GWO) [27, 28], salp swarm algorithm (SSA) [29] and grasshopper optimization algorithm (GOA) [30] have proven very successful for power optimization under partial shading conditions (PSCs) without knowing the shapes of the PV characteristic curves. These approaches are quite good in tracking the global MPP (GMPP); however, they have a long tracking time and startup oscillations [23]. As a result, various improvements to traditional SITs have been made to address these issues. The GOA is a viable option to track MPP in PV systems because there is only one parameter that needs to be managed. The fundamental flaw in this method is the high startup oscillations during GMPP tracking in PV systems, which significantly reduces the energy harvested [7].

The GOA was created in 2017 which mimics the grasshopper behavior during the food-search process [31]. In 2020, it was used for the first time to track the MPP of a PV system [30]. The authors compared its result to those of dragonfly optimization (DFO), artificial bee colony (ABC), PSO-based gravitational search (PSOGS), PSO, and CSA under complex PSC (CPSC). However, it had startup oscillations, a longer settling time of 0.39 s, and a tracking time of 0.16 s. A hybrid algorithm between the GOA and FLC was introduced in 2020 which combined the benefits of the GOA and the FLC [32]. FLC's membership functions were fine-tuned by GOA to account for all the uncertainties resulting from varying irradiance and temperature. It outperformed the P&O and FLC methods in terms of convergence speed and tracking efficiency with increased system complexity and oscillations. Furthermore, it did not consider the PSC in work. A modified GOA combined with an incremental conductance (INC) approach was published in 2020 [33]. It was used in the initial part of the GMPP tracking process and the system then shifted to the second stage by employing INC to obtain the proper GMPP. It tracked the global peaks more effectively than PSO and the modified firefly algorithm (MFA) at the cost of additional settling time 1.97 s, tracking time 1.87 s, and undesirable oscillations. In 2020, another GOA-tuned MPP strategy was introduced by [34] to obtain an optimum duty cycle for a boost converter. The system was tested with start-up transients, line disturbances, load disturbances, servo settings,

and PSCs. It outperformed the P&O and PSO in terms of rise time and settling time. However, the authors discussed the proposed technique's behavior for limited peaks without mentioning the swarm size. In 2021, a study was conducted to increase the GOA's performance [35]. The proposed strategy was compared to P&O, PSO, and differential evolution in terms of power extraction and conversion efficiency. The findings show that the proposed method outperformed its competitors. However, the tracking time 2.95 s and settling time 3.10 s have been increased to seconds.

In this research work, an improved GOA (IGOA) is presented to track a GMPP in a PV system under PSCs that addresses flaws such as high tracking time and startup oscillations. The duty ratio of the boost converter in the system is taken as a grasshopper (GH), and a normalized irradiance factor is added to balance the forces of attraction and repulsion between the GHs while maintaining simple logic and structure. A MATLAB/Simulink model is developed to evaluate the performance of the IGOA under uniform irradiance, 4 peaks, 5 peaks, and 6 peaks to reduce tracking time and startup oscillations. The simulation is run 100 times to ensure the consistency of the proposed algorithm in tracking a GMPP. The results are compared with well-known SI optimization techniques such as GOA, grey wolf optimization (GWO), cuckoo search algorithm (CSA), and salp swarm algorithm (SSA) under different shadowing conditions in terms of tracking time, settling time, tracking failure rate (FR), and startup oscillations. The superiority of IGOA over competitor techniques is demonstrated by detailed statistical analysis. Comparative results and statistical analysis show a clear advantage of IGOA over its counterparts that exhibit faster tracking speed, shorter settling time, reduced startup oscillations, and lower FR. IGOA guarantees a tracking time of 0.07-0.15 s, which is 2 to 24 times faster than other SITs, allowing quick adaptability to changes in atmospheric conditions. Unlike other SITs, it shows a settling time of 0.08-0.17 s which is 6 to 13 times shorter, and shows that varying the number of peaks due to shading conditions has no impact on settling time for a specific swarm size. It ensures high reliability in MPP tracking with less than 2% FR in the 3 peaks system for all swarms and 0% FR for the rest of the setup. This unique characteristic highlights the robustness of the IGOA and distinguishes it from other SITs. Increasing the swarm agents leads to a reduction in FR of IGOA, which highlights its scalability and ensures a higher possibility of finding acceptable solutions. The remaining paper is arranged as follows. The effect of PSCs on the PV system is described in Section 2. Section 3 explores the proposed method. The simulation setup is presented in Section 4. Section 5 presents the results and Section 6 is reserved for the conclusion.

## 2 Effect of PSC on PV system

The PV array or even a single module may receive nonuniform irradiation due to the shade of adjacent buildings, dust, clouds, trees and bird litter, which can completely or partially cover the PV module/array [36]. Fig 1 shows the unshaded PV array, shaded PV array, and their characteristic curves. The module/array undergoes a faster and more dynamic solar insolation shift during cloudy days, which has a direct impact on its performance [37]. Under PSCs, unshaded modules receive a certain amount of solar irradiation, while shaded modules receive less than that amount [38]. This imbalance in irradiance causes a hotspot in the module that receives less irradiance [39]. This module operates as a load on the system which may be damaged as a result of hotspots. The output power of the system could be as low as 70% of its total installed capacity which in turn lowers the system's efficiency [40]. To protect PV modules from self-heating and to increase efficiency during PSC, bypass diodes are connected to the modules to offer a different current path [5, 41]. The characteristic curves exhibit a combination of LP and a global peak (GP) due to the insertion of diodes. The offline and online MPPT

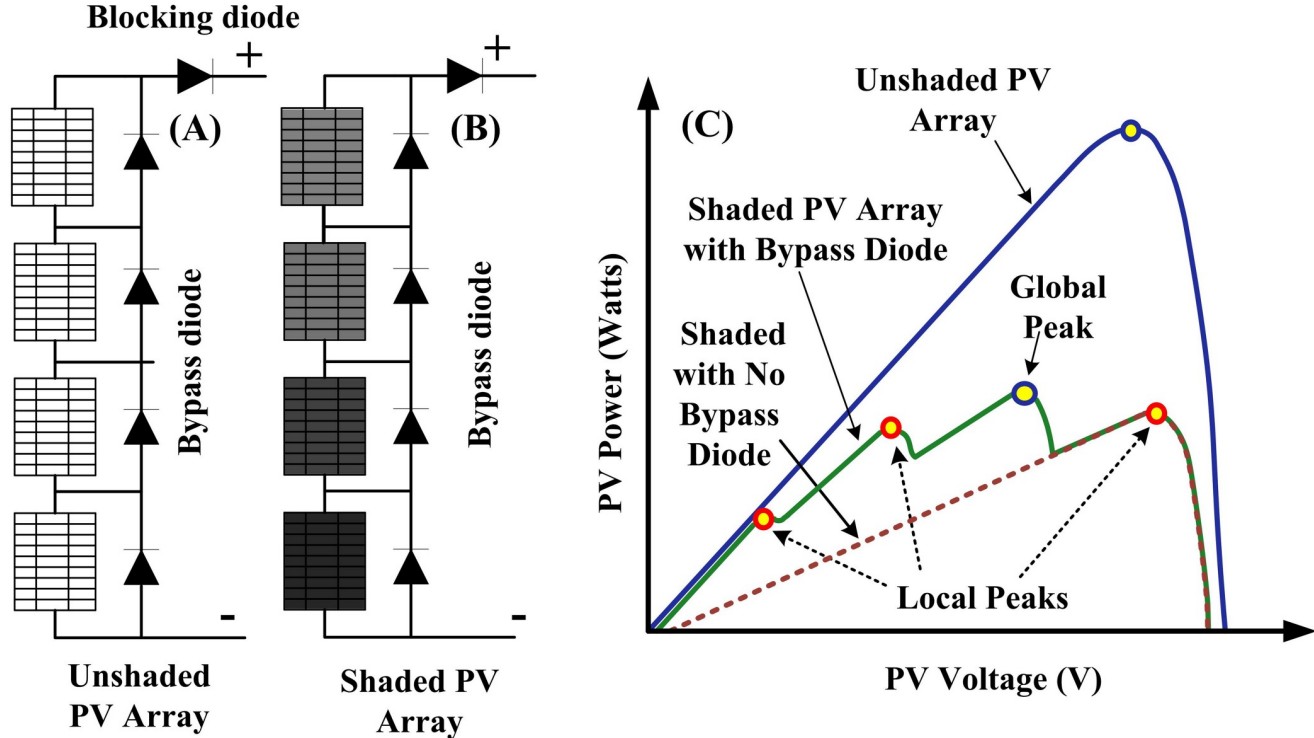

**Fig 1. Unshaded and shaded PV array.**

algorithms cannot differentiate a GP from many LPs [42]. Therefore, a strong MPPT approach is required capable of tracking a GP surrounded by numerous local peaks [43].

## 3 Grasshopper optimization algorithm

The GOA was created in 2017 which mimics the GH behavior during the food-search process [31]. In 2020, it was used for the first time to track the MPP of a PV system [30]. It is a modern nature-inspired swarm-based algorithm that has two stages in the search process. The nymph is the first stage that does not have wings. Its slow movement in small steps is termed exploitation [31]. The adult is the second stage, having wings that move faster in the air at random intervals to avoid predators and search for food in a larger search area. This abrupt and fast movement of the adult stage in a larger search region is termed exploration. The region where exploitation and exploration are equal is called the comfort zone. Both the nymph and the adult grasshoppers have swarming behavior. The mathematical form of the position $X_i$ of a GH is given by (1) [44].

$$X_i = \sum_{j=1, j \neq i}^{N} s(\|x_j - x_i\|) \frac{x_j - x_i}{D_{ij}} - g\hat{e}_g + u\hat{e}_w \qquad (1)$$

Where $N$ is the number of GHs, $D_{ij}$ is the displacement between two GHs $i$ and $j$ given in (5), $g$ is the gravitational constant and $\hat{e}_g$ is its unit vector, $u$ is the drift constant caused by the wind with its unit vector $\hat{e}_w$ [45]. $s$ represents the force of attraction and repulsion between the GHs

at a distance $r$ given in (2).

$$s(r) = f \exp^{\left(\frac{-r}{l}\right)} - \exp^{-r} \tag{2}$$

Here $f$ represents attraction intensity and $l$ is a scale of attractive length. GHs stop exploring the search space as (1) quickly converges toward the comfort zone that is not required. To solve the issue, it is assumed that the wind is always towards the target and gravity is always towards the center [46]. Therefore, (1) can be modified to (3) ignoring the terms gravitational force and wind advection.

$$X_i = c \sum_{j=1, j \neq i}^{N} c \left( \frac{U_l - L_l}{2} s(\|x_j - x_i\|) \frac{x_j - x_i}{D_{ij}} \right) + \hat{T}_d \tag{3}$$

$U_l$, $L_l$ are the higher and lower limits of the search space, $\hat{T}_d$ is the target position in the $d_{th}$ dimension, and $c$ is a controlling parameter in (4) which controls the comfort zone, repulsive and attractive zone.

$$c = c_u - l \frac{c_u - c_l}{L} \tag{4}$$

$c_u$, $c_l$ are the upper and lower limits of $c$, $l$ is the present iteration and $L$ is the total number of iterations. To calculate the distance between the two GHs, a fully vectorized Euclidean distance formula (5) is applied. The distance between the two matrices containing various GH positions say **A** and **B** is **D** [31].

$$D = \|A - B\| = \sqrt{\|A\|^2 + \|B\|^2 - 2A.B} \tag{5}$$

## 3.1 Proposed improved grasshopper optimization algorithm

The impact of social interaction between the two GHs in the distance range [0, 10] is given in [46]. It can be seen that the two GHs repel each other when they are close enough in the distance [0,2.079] units. Beyond this, the force of attraction increases up to a distance of 4 units, then it decreases gradually. The point where the two forces are equal is called the comfort zone which is about 2.079 units away from the other GH. These distance values are calculated when the intensity of the interaction $f$ is 0.5 and the length scale $l$ is 1.5 [31]. Most of the GOA-based MPPT published research work normalizes the distance between the GHs in the range [0, 4]. However, if the duty ratio $d$ of the converter is taken as a position of a GH then the maximum range of duty ratio, the distance between the two GHs, lies between [0, 1]. Therefore, the social interaction between GHs should be reconsidered in this range. Fig 2a shows that when $f$ is 0.5 for all values of $l$ in the range [1, 3], there is no force of attraction and no comfort zone (CZ); only the force of repulsion is present. On the contrary, when $f$ is 1, just the attraction force is present and there is no comfort zone or repulsion force for any value of $l$. GHs can explore the search space using repulsion forces and can exploit the target location using attraction forces [34]. This means that exploration is easy in the former case, which makes converging at a point very difficult as there is no attraction force. Exploitation is easy in the latter case, which makes a larger search area to scan. There should be a balance between the two forces. It is evident from Fig 2 that we get the best equilibrium between the attraction and repulsion forces when $f$ is between [0.65,0.9] and $l$ is in the range [2,2.9]. Therefore, it is necessary to redefine the range of $f$ and $l$ in the distance range [0, 1] as the maximum distance between the two GH (duty ratio) is 1. Therefore, social interaction ('$s$' function) is modified to (6) to balance the

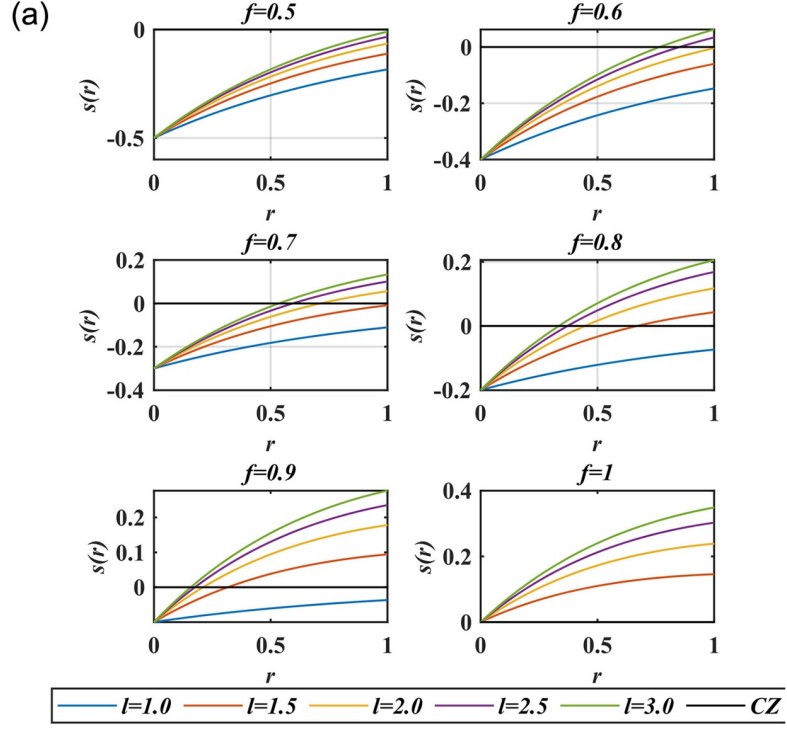

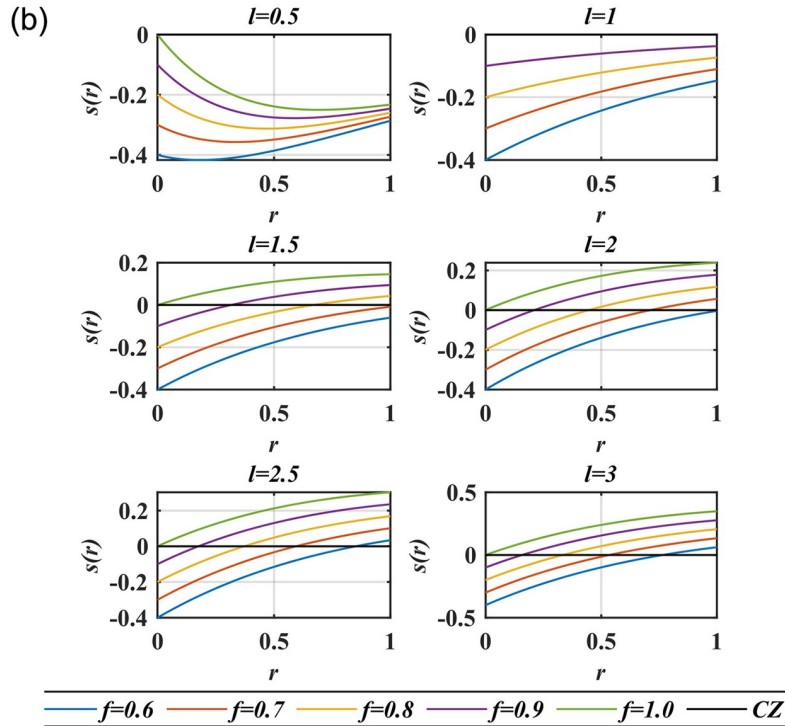

**Fig 2. Social interaction for varying (a) *l*, (b) *f* in distance range [0, 1].**

two forces.

$$s(r) = f_{new}\exp^{\left(\frac{-r}{l_{new}}\right)} - \exp^{-r} \tag{6}$$

Here $f_{new}$ and $l_{new}$ are defined in (7) and (8) respectively, while $f$ and $l$ retain their previous values, that is, 0.5 and 1.5, respectively. Moreover, for intermittent environmental conditions a normalized average irradiance factor $G_N$ is added in the equations to get $f_{new}$ in between [0.65,0.9] and $l_{new}$ in the range [2,2.9].

$$f_{new} = f(G_N + x) \tag{7}$$

$$l_{new} = l(G_N + x) \tag{8}$$

Where $x$ is from one of the following two cases.

Case 1: if $(0 < G_N \leq 0.5)$ then x = 1.3

Case 2: if $(0.5 < G_N \leq 1)$ then x = 0.8

The position equation becomes 9 in which $d_i$ is the position of $i_{th}$ GH and $d_u$ and $d_l$ represent its upper and lower limits respectively.

$$d_i = c \sum_{j=1,j\neq i}^{N} c\left(\frac{d_u - d_l}{2} s(\|d_j - d_i\|)\frac{d_j - d_i}{D_{ij}}\right) + \hat{T}_d \tag{9}$$

The objective function is defined as (10).

$$P(d_i^{(l)}) > P(d_i^{(l-1)}) \tag{10}$$

Where $P$ is the power received for a specific $d_i$ at iteration number $l$ for the $i_{th}$ GH in IGOA. Here, $d_i$ is the position of $i^{th}$ GH that is to be optimized. The algorithm aims to find the optimal $d$ that maximizes the power received by the system. At each iteration, the power received for a specific $d_i^{(l)}$ is evaluated. This power value represents the fitness or objective value of the GH. The algorithm compares this power value in the current iteration $l$ with the power value in the previous iteration $l - 1$. The inequality in (10) serves as a criterion to determine whether the GHs have improved their positions ($d$) in terms of maximizing the power received. If the power received in the current iteration is greater than the power received in the previous iteration, this implies that the GHs have made progress and have moved towards a more optimal $d$. Repetition of this process over several iterations is key to moving GHs towards the optimal $d$ and thus reaching the maximum output power target.

Flow chart of the IGOA is shown in Fig 3. Initially, the IGOA initializes all its parameters and randomly generates the initial positions ($d_i$) of each GH within the limits. After that, it calculates the fitness value (output power) of each GH based on the objective function (10). Then it identifies and stores the best fitness value (highest output power) and the corresponding $d$ as the target position. After this process, the iterative cycle starts and updates the variables with their respective equations as depicted in the flow chart. Then it recalculates the fitness values for the updated positions and compares this with the best fitness value obtained so far. If any GH has a better fitness value, it updates the best fitness value accordingly. It completes its iterative cycle using the equations specified in the flow chart. When the current iteration reaches its maximum limit, it checks the change in irradiance. If there is a change, it repeats the whole process otherwise it terminates. After termination, it returns the best target position which

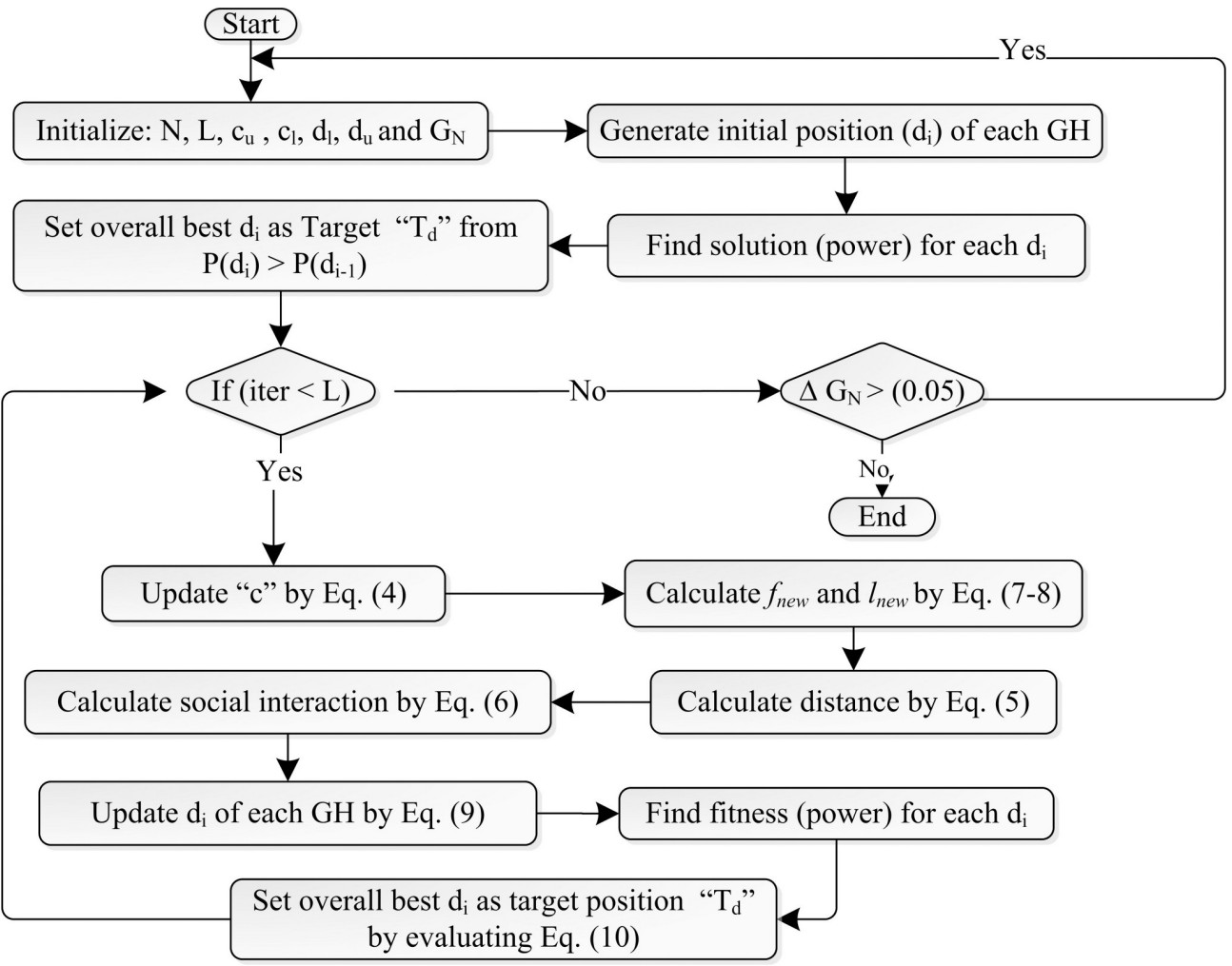

**Fig 3. Flow chart of the proposed IGOA.**

represents the optimized $d$ that satisfies the objective function (10) with maximum output power.

## 4 Simulation setup

A simple model is taken to evaluate the viability of algorithms under various PSCs, as shown in Fig 4. The PV array receives different levels of irradiance (G) from the visible light spectrum, which results in current and, ultimately, electricity [47, 48]. The generated power enters a boost converter, which modifies the output voltage according to the DC link voltage level. A tiny capacitor (C) is inserted in between the PV array and the boost converter to supply the ripple current required for opening and closing the converter's MOSFET. SITs are used one by one to monitor PV current and voltage to track the system's MPP. Sun-Power PV modules (SPR-315E-WHT-D) are connected in series to make an array. The performance of the system is evaluated for one unshaded condition and three PSCs provided in Table 1 that have 4 to 6 peaks in their characteristic curves shown in Fig 5. All peaks are evaluated for various numbers of swarms ranging from 3 to 6. The simulation is run 100 times to combat the unpredictable behavior of SITs.

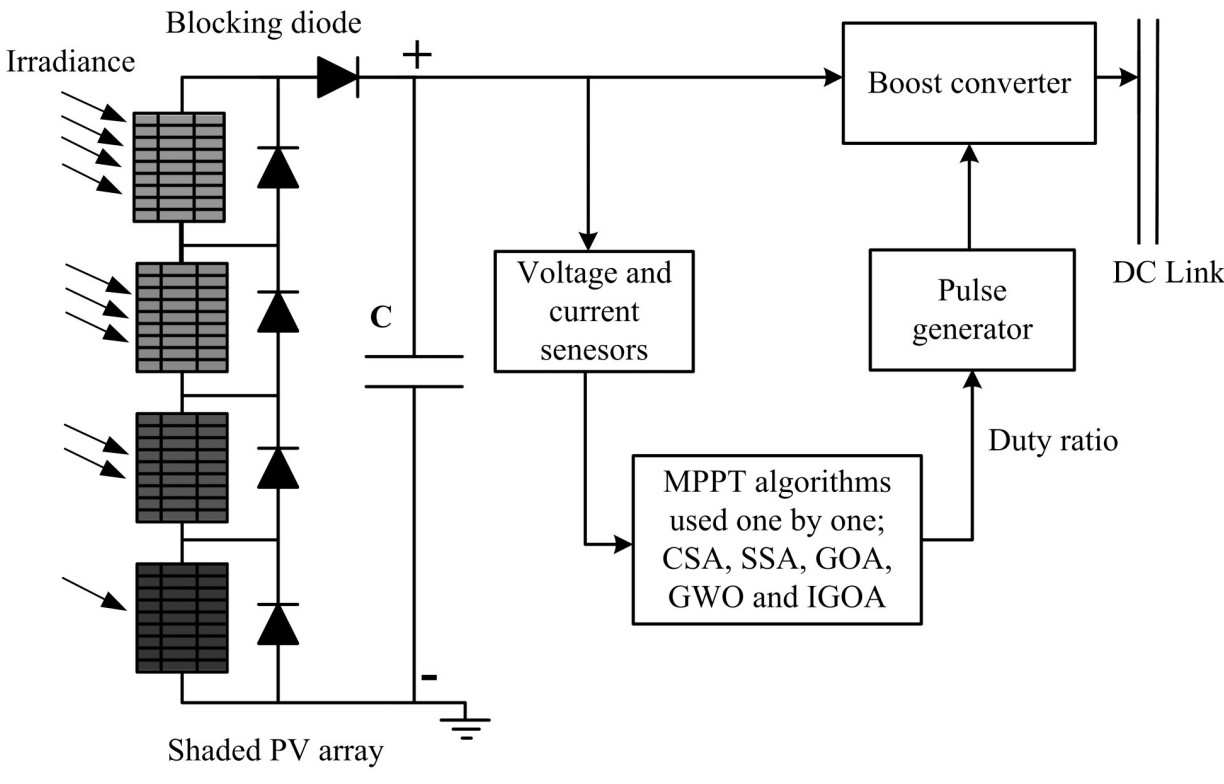

**Fig 4. Block diagram of the Simulink model.**

## 5 Results and discussion

The proposed IGOA is compared with four other SITs: CSA, GOA, GWO and SSA in terms of tracking and settling times, failure rate (FR) and startup oscillations. The FR is stated as the ratio between the number of attempts maturing at local peaks to the total number of attempts during the simulations. Some of the algorithms may perform well for fewer peaks (fewer shading conditions), and others may perform well under more shading conditions. Similarly, the number of swarms may also affect the performance of the system. Therefore, for a fair comparison, these techniques are studied for 1, 4, 5, and 6 peaks with swarm sizes of 3, 4, 5, and 6. Table 2 depicts the comparison results. The following subsections present the comparative performance analysis of SITs for various peaks and swarm numbers.

### 5.1 Comparison of SITs for four peaks with multiple agents

In this section, the comparison among SITs is performed for four peaks and 3, 4, 5, and 6 swarm agents. In the case of four peaks and three swarm agents (4P3S), the GWO tracking

**Table 1. Irradiance level on each PV module ($W/m^2$) and GMPP.**

|  | PV-1 | PV-2 | PV-3 | PV-4 | PV-5 | PV-6 | GMPP (W) |
|---|---|---|---|---|---|---|---|
| Case-1 | 800 | 250 | 700 | 400 | - | - | 450 |
| Case-2 | 100 | 500 | 700 | 800 | 600 | - | 680 |
| Case-3 | 750 | 500 | 900 | 800 | 400 | 1000 | 999 |
| Case-4 | 1000 | 1000 | 1000 | 1000 | - | - | 1260 |

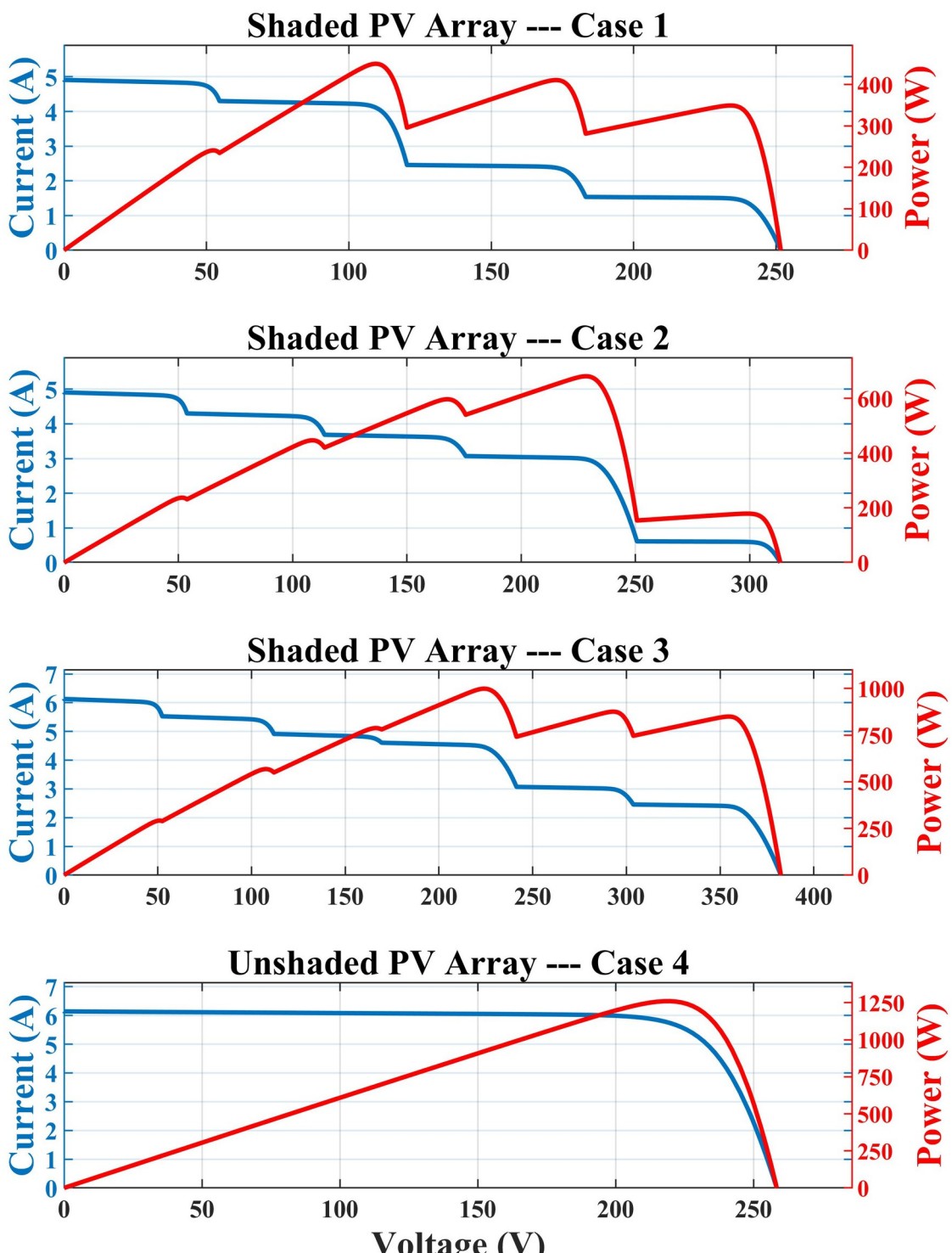

**Fig 5. PV Characteristics curves under shaded and unshaded conditions.**

**Table 2. Simulink results comparison of SITs.**

| Algo | NS | Unshaded: 1 Peak | | shaded PV array: 4 Peaks | | | | shaded PV array: 5 Peaks | | | | shaded PV array: 6 Peaks | | | | Average of shaded PV | | |
|---|---|---|---|---|---|---|---|---|---|---|---|---|---|---|---|---|---|---|
| | | $t_t$ (s) | $t_s$ (s) | $t_t$ (s) | $t_s$ (s) | FR | SOs | $t_t$ (s) | $t_s$ (s) | FR | SOs | $t_t$ (s) | $t_s$ (s) | FR | SOs | $t_t$ (s) | $t_s$ (s) | FR |
| IGOA | 3 | 0.04 | 0.05 | 0.07 | 0.12 | 1 | VL | 0.08 | 0.13 | 2 | VL | 0.07 | 0.11 | 3 | VL | 0.07 | 0.12 | 2.00 |
| GOA | | 0.35 | 0.42 | 0.75 | 0.99 | 6 | M | 0.45 | 0.76 | 8 | L | 0.35 | 0.63 | 12 | L | 0.52 | 0.79 | 8.67 |
| SSA | | 0.31 | 0.39 | 0.68 | 0.99 | 6 | L | 0.53 | 0.81 | 9 | L | 0.48 | 0.78 | 14 | L | 0.56 | 0.86 | 9.67 |
| GWO | | 0.67 | 0.75 | 1.02 | 1.57 | 7 | L | 0.89 | 1.23 | 16 | L | 0.63 | 0.95 | 23 | L | 0.85 | 1.25 | 15.3 |
| CSA | | 0.17 | 0.23 | 0.89 | 1.52 | 9 | M | 1.22 | 1.75 | 13 | H | 1.23 | 1.53 | 19 | M | 1.11 | 1.60 | 13.67 |
| IGOA | 4 | 0.04 | 0.05 | 0.09 | 0.12 | 0 | VL | 0.1 | 0.13 | 0 | VL | 0.11 | 0.13 | 0 | VL | 0.10 | 0.13 | 0.00 |
| GOA | | 0.47 | 0.53 | 0.59 | 0.87 | 1 | M | 0.42 | 0.61 | 1 | L | 0.41 | 0.67 | 3 | L | 0.47 | 0.72 | 1.67 |
| SSA | | 0.38 | 0.43 | 0.67 | 1.02 | 1 | M | 0.95 | 1.31 | 3 | L | 0.63 | 1.12 | 4 | L | 0.75 | 1.15 | 2.67 |
| GWO | | 0.76 | 0.83 | 1.39 | 1.99 | 2 | VH | 0.99 | 1.32 | 5 | M | 0.71 | 0.99 | 7 | M | 1.03 | 1.43 | 4.67 |
| CSA | | 0.26 | 0.32 | 2.19 | 2.57 | 2 | VH | 0.65 | 0.83 | 4 | M | 1.32 | 1.61 | 7 | M | 1.39 | 1.67 | 4.33 |
| IGOA | 5 | 0.05 | 0.06 | 0.12 | 0.14 | 0 | VL | 0.13 | 0.17 | 0 | VL | 0.12 | 0.19 | 0 | VL | 0.12 | 0.17 | 0.00 |
| GOA | | 0.56 | 0.64 | 0.69 | 1.77 | 0 | M | 1.02 | 1.24 | 0 | L | 0.35 | 0.52 | 2 | L | 0.69 | 1.18 | 0.67 |
| SSA | | 0.47 | 0.59 | 1.89 | 2.21 | 0 | VH | 1.69 | 2.03 | 0 | L | 1.32 | 1.69 | 4 | L | 1.63 | 1.98 | 1.33 |
| GWO | | 0.87 | 0.92 | 1.57 | 1.95 | 0 | VH | 0.61 | 1.34 | 2 | H | 0.73 | 1.09 | 4 | M | 0.97 | 1.46 | 2.00 |
| CSA | | 0.32 | 0.37 | 1.95 | 2.28 | 0 | VH | 1.31 | 1.59 | 3 | H | 1.41 | 1.79 | 5 | H | 1.56 | 1.89 | 2.67 |
| IGOA | 6 | 0.06 | 0.08 | 0.15 | 0.18 | 0 | VL | 0.14 | 0.18 | 0 | VL | 0.14 | 0.17 | 0 | VL | 0.14 | 0.18 | 0.00 |
| GOA | | 0.61 | 0.72 | 0.66 | 1.01 | 0 | M | 0.39 | 0.54 | 0 | L | 0.29 | 1.63 | 0 | L | 0.45 | 1.06 | 0.00 |
| SSA | | 0.57 | 0.63 | 1.81 | 1.99 | 0 | VH | 1.49 | 1.65 | 0 | H | 0.97 | 1.89 | 0 | VH | 1.42 | 1.84 | 0.00 |
| GWO | | 0.99 | 1.18 | 1.72 | 1.84 | 0 | M | 1.37 | 1.51 | 1 | VL | 1.53 | 1.85 | 1 | VL | 1.54 | 1.73 | 0.67 |
| CSA | | 0.37 | 0.41 | 2.49 | 3.2 | 1 | VH | 1.39 | 1.76 | 1 | H | 1.32 | 1.66 | 2 | M | 1.73 | 2.21 | 1.33 |

Algo: Algorithm, NS: No. of Swarms, $t_t$: Average Tracking time, $t_s$: Average Settling Time, *FR*: Failure Rate (%)

SOs: Startup Oscillations, VL: Very Low, L: Low, M: Medium, H: High, VH: Very High

time is 1.02 s, the settling time is 1.57 s, and 7% FR. The other SITs' tracking time is more than 0.6 s except for IGOA which takes only 0.07 s for tracking and 0.12 s for settling time with 1% FR and very low startup oscillations. SSA and GOA spent 0.99 s in settling time with 6% FR. CSA exhibits more oscillations than the other SITs. With the 4P4S setting, IGOA tracked GMPP in 0.09 s with 0% FR; all other SITs showed longer tracking and settling times with more than 1% FR. The startup oscillations are high in the case of CSA and GWO. Moreover, the CSA tracking time is 2.19 s which is high among all the other SITs used in the study. With 4P5S, all SITs had 0% FR but very high oscillations, except IGOA, which showed very low oscillations. IGOA tracking time is 0.12 s, compared to 1.95 s for CSA, 1.89 s for SSA, 1.57 s for GWO, and 0.69 s for GOA. When the swarm setting was changed to 4P6S, all SITs maintained 0% FR, but with an increase in oscillations except IGOA. Fig 6a shows the fitness value (power obtained) over time and Fig 6b depicts the graph between time and duty ratios for the 4P6S setup. It is evident that the duty rate for IGOA exhibits a rapid increase and tracks the GMPP within 0.078 s and settles the output in 0.15 s. This indicates that IGOA achieves fitness values faster than the other algorithms as time progresses. On the other hand, other algorithms show more oscillations and more time for convergence. Compared to IGOA, their duty ratios take longer to find the best solution. The CSA tracking time is the longest, due to the random nature of the inherent Lévy flight. It is, therefore, recommended to use at least four swarm agents with IGOA and five swarm agents with other SITs when the PV characteristic curve has up to four peaks. It is evident that the tracking times, settling times, and FR for IGOA are lower than the other SITs with swarms 3 to 6 for 4 Peaks.

(a)

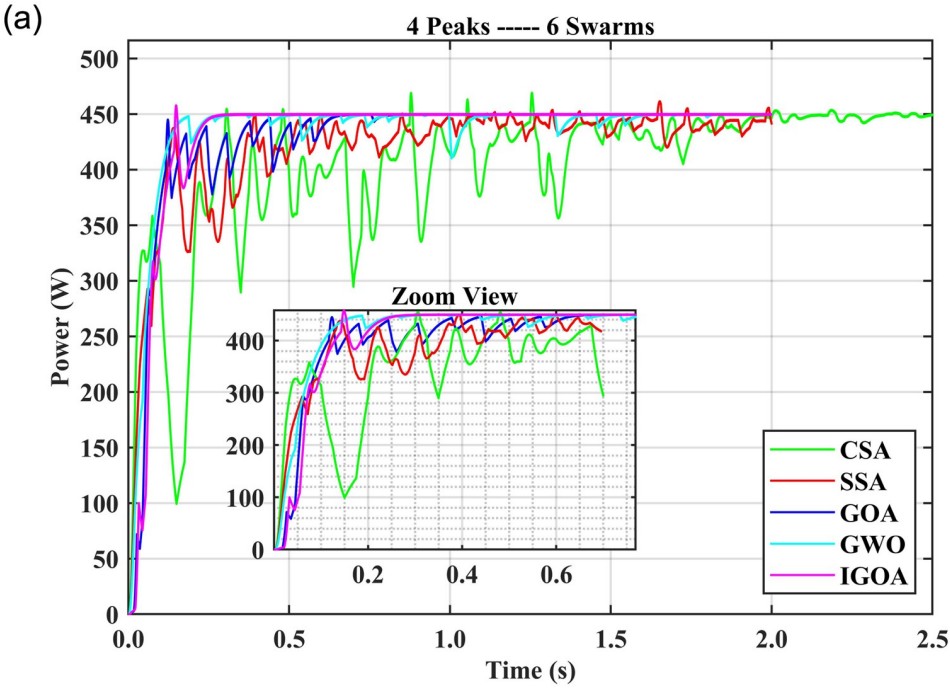

(b)

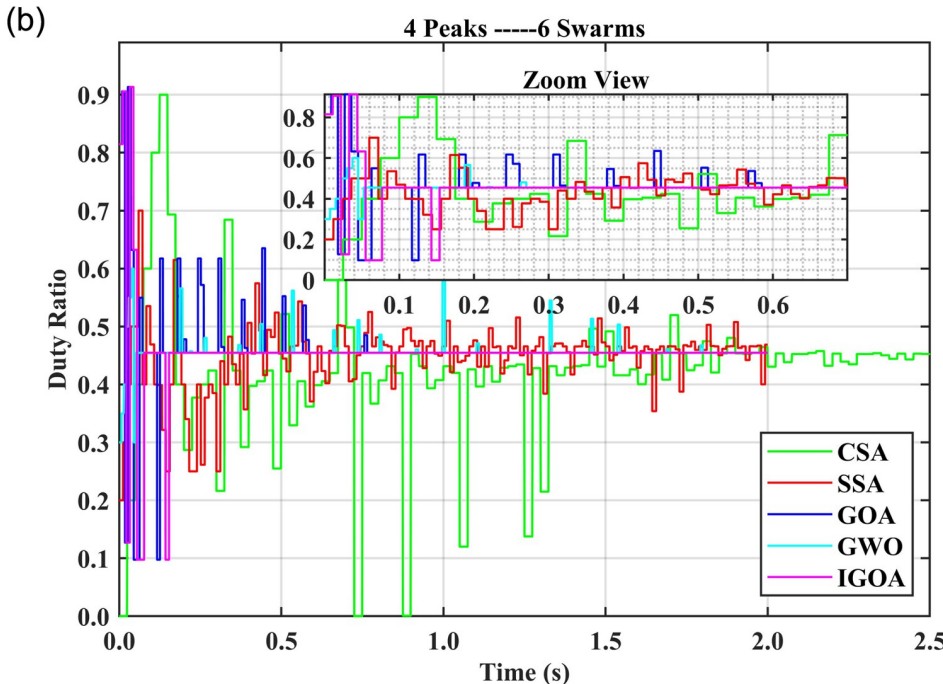

**Fig 6. (a) Output powers for the 4P6S setup, (b) duty ratios for the 4P6S setup.**

## 5.2 Comparison of SITs for five peaks with multiple agents

In the case of five peaks and three swarm agents (5P3S), the CSA tracking time is 1.22 s and the settling time is 1.55 s with 13% FR and GWO takes 0.89 s in the tracking and 1.23 s for the settling time with 16% FR. IGOA spent only 0.08 s in tracking and 0.13 s for settling time with 2% FR and very low startup oscillations. SSA and GOA took more than 0.45 s in tracking and

0.76 s in settling time with 9% FR. The CSA exhibits more oscillations than the other SITs used. With 5P4S, IGOA tracked GP in 0.1 s with 0% FR; all other SIT showed longer tracking and settling times with more than 3% FR. The startup oscillations are high in the case of GWO. Furthermore, the settlement time of GWO and SSA is more than 1.3 s which is high among all other SITs used in this case. With 5P5S, all SIT had 0% FR except CSA with 3% FR. IGOA tracking time is 0.13 s, compared to 1.69 s for SSA, 1.02 s for GOA, and 1.31 s for CSA. SSA and CSA show high oscillations as compared to others. For six swarms (5P6S) all the SITs show 0% FR except CSA and GWO which still have 1% FR. Fig 7a shows the fitness value (power obtained) over time and Fig 7b depicts the graph between time and duty ratios for the 5P6S setup. It is evident that the duty rate for IGOA exhibits a rapid increase and tracks the GMPP within 0.07 s and settles the output in 0.14 s. On the other hand, tracking time for SSA is 1.49 s, 1.37 s for GWO, 1.39 s for CSA, and 0.39 s for GOA and show more oscillations. This indicates that IGOA achieves fitness values faster than the other algorithms as time progresses. Compared to IGOA, their duty ratios take longer to find the best solution. It is therefore recommended to use at least four swarms with IGOA, 5 swarms for GOA and SSA, and six swarms for other SIT when the PV characteristic curve has up to five peaks. Furthermore, the tracking time for IGOA is lower than 0.14 s for swarms up to six. It can be observed that IGOA maintains very low oscillations for all the peaks and swarm sizes discussed so far.

## 5.3 Comparison of SITs for six peaks with multiple agents

In the case of six peaks and three swarm agents (6P3S), the tracking time for CSA is 1.23 s, 0.63 s for GWO, 0.48 s for SSA and 0.35 s for GOA with 19%, 23%, 14%, and 12% FR respectively. IGOA spent only 0.07s in tracking and 0.11 s in settling time with a 3% failure rate and very low startup oscillations. With 6P4S, IGOA tracked the GMPP in 0.11 s with 0% FR; all the other SIT showed both longer tracking and settling times. With 6P5S, only IGOA has 0% FR with very low oscillations. IGOA tracking time is 0.12 s, compared to 1.41 s for CSA, 1.32 s for SSA, 0.73 s for GWO, and 0.35 s for GOA. For 6P6S SSA, GOA and IGOA maintained 0% FR. Fig 8a shows the fitness value (power obtained) over time and Fig 8b depicts the graph between time and duty ratios for the 6P6S setup. It is evident that the duty rate for IGOA exhibits a rapid increase and tracks the GMPP within 0.07 s and settles the output in 0.16 s. On the other hand, settling time for CSA is 1.66 s, GWO 1.85 s, SSA 1.89 s, and 1.63 s for GOA. This indicates that IGOA achieves fitness values faster than the other algorithms as time progresses. Compared to IGOA, their duty ratios take longer to find the best solution. It is therefore recommended to use at least four swarms with IGOA and six swarms with GOA and SSA and more than 6 swarms with other SITs when the PV characteristic curve has up to six peaks. Moreover, the tracking time for IGOA is less than 0.14 s for swarm sizes of up to six. It is evident that the proposed IGOA technique demonstrated its supremacy by tracking the GMP up to 6-10 times faster than the compared algorithms.

## 5.4 Comparison of SITs for single peak with multiple agents

In the case of a uniform PV array with 1P3S configuration, the GWO tracking time is 0.67 s, the settling time is 0.75 s, and 0% FR. The other SITs' tracking time is more than 0.3 s except for IGOA which takes only 0.04 s for tracking and 0.05 s for settling time without startup oscillations and 0% FR. CSA exhibits more oscillations than the other SITs. With the 1P4S setting, IGOA tracked GMPP in 0.04 s with 0% FR; all other SITs showed longer tracking and settling times. The startup oscillations are high in the case of CSA and GWO. In addition, the GWO tracking time is 0.76 s which is high among all other SITs used in the study. With 1P5S, all SITs had 0% FR. When the swarm setting changed to 1P6S, all SITs

(a)

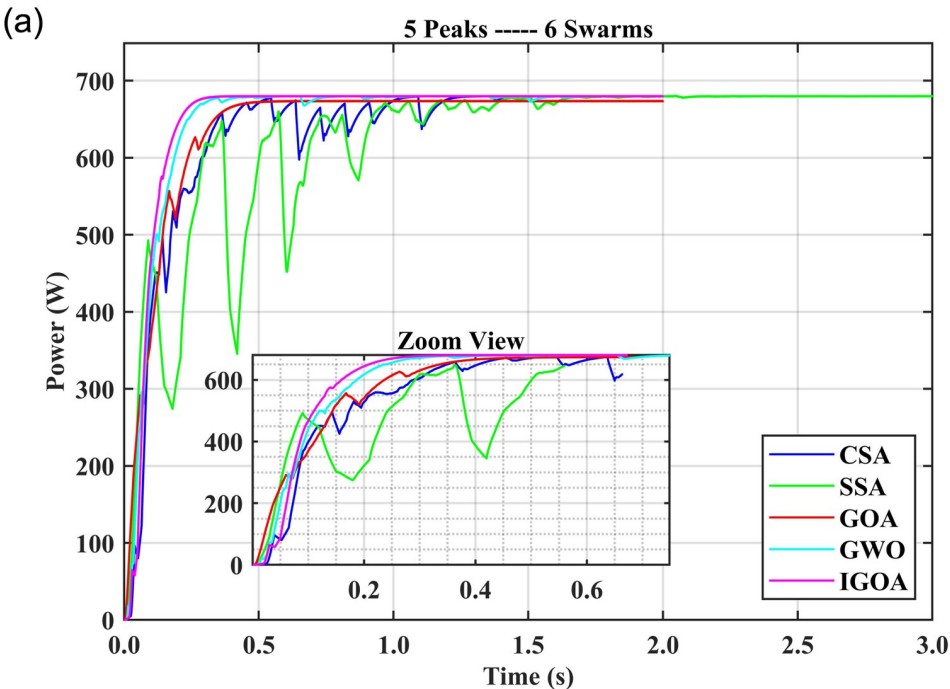

(b)

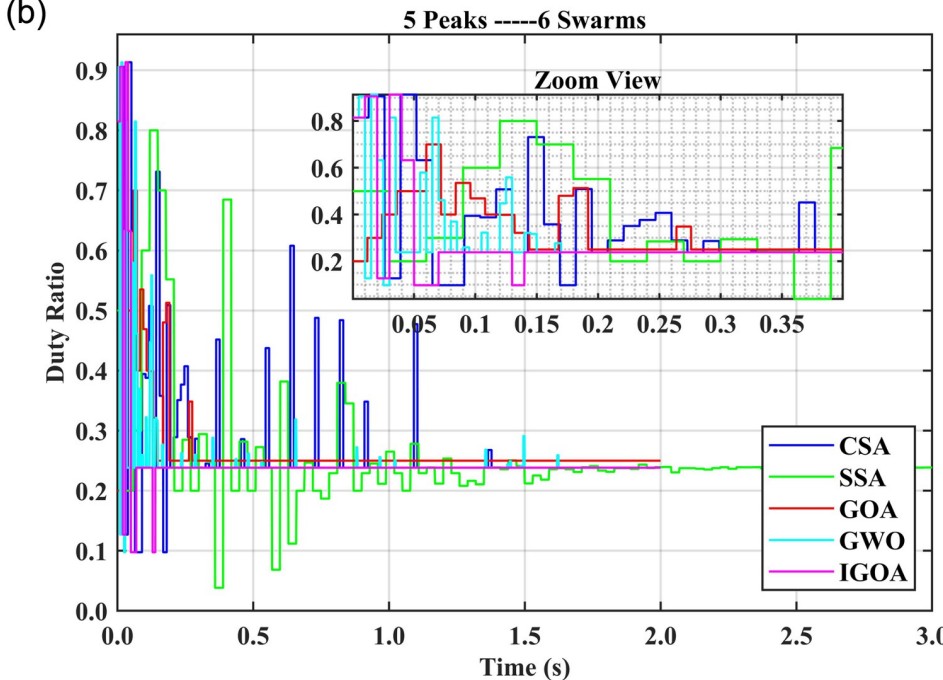

**Fig 7. (a) output powers for the 5P6S setup and (b) duty ratios for the 5P6S setup.**

maintained 0% FR. Fig 9a shows the fitness value (power obtained) over time and Fig 9b depicts the graph between time and duty ratios for the 1P6S setup. It is evident that the duty rate for IGOA exhibits a rapid increase and tracks the GMPP within 0.03 s and settles the output in 0.07 s. This indicates that IGOA achieves fitness values faster than the other

(a)

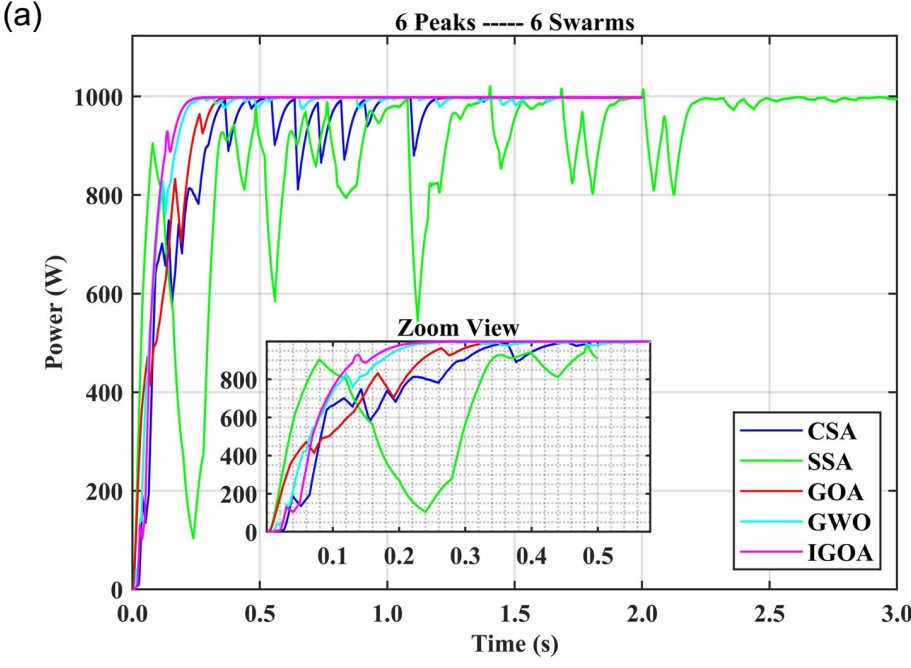

(b)

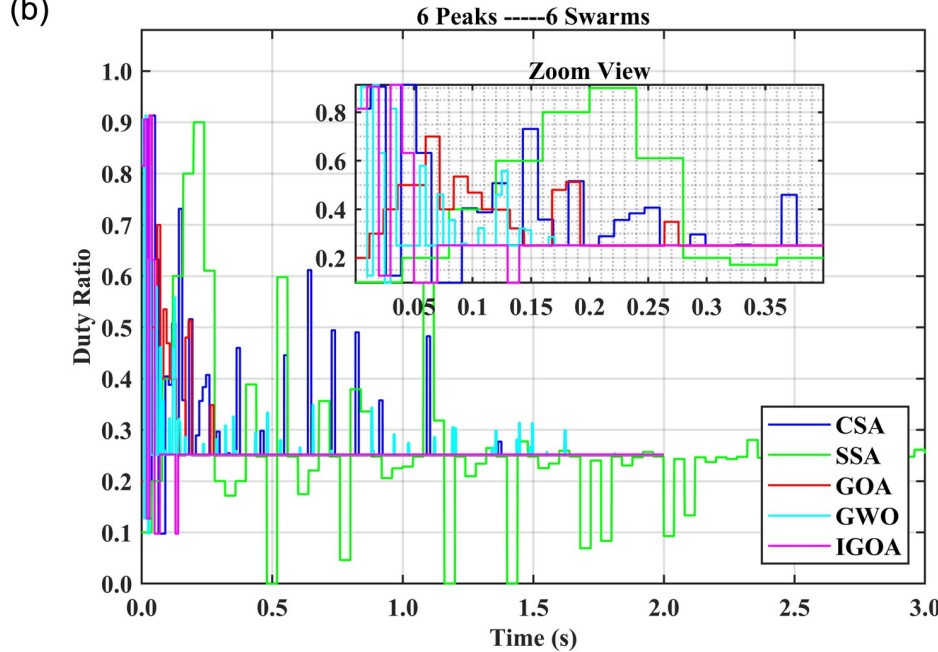

**Fig 8. (a) Output powers for 6P6S setup, (b) duty ratios for 6P6S setup.**

algorithms as time progresses. On the other hand, other algorithms show more oscillations and more time for convergence. Compared to IGOA, their duty ratios take longer to find the best solution. The CSA tracking time is the longest, due to the random nature of the inherent Lévy flight. Still, IGOA is the fastest without oscillations. It is evident that the tracking times, settling times, and FR for IGOA are lower than the other SITs with swarms 3 to 6. The

(a)
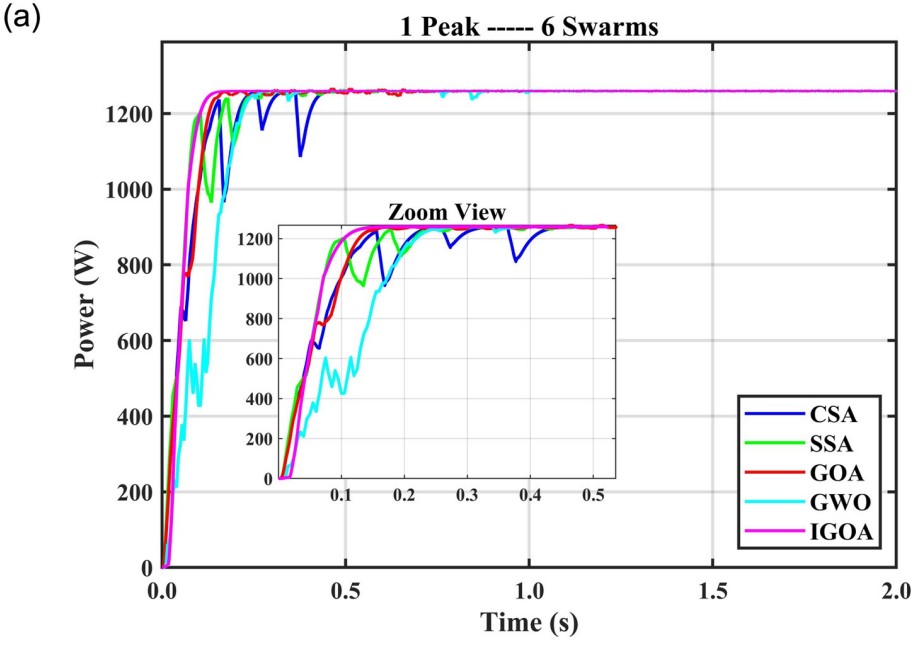

(b)
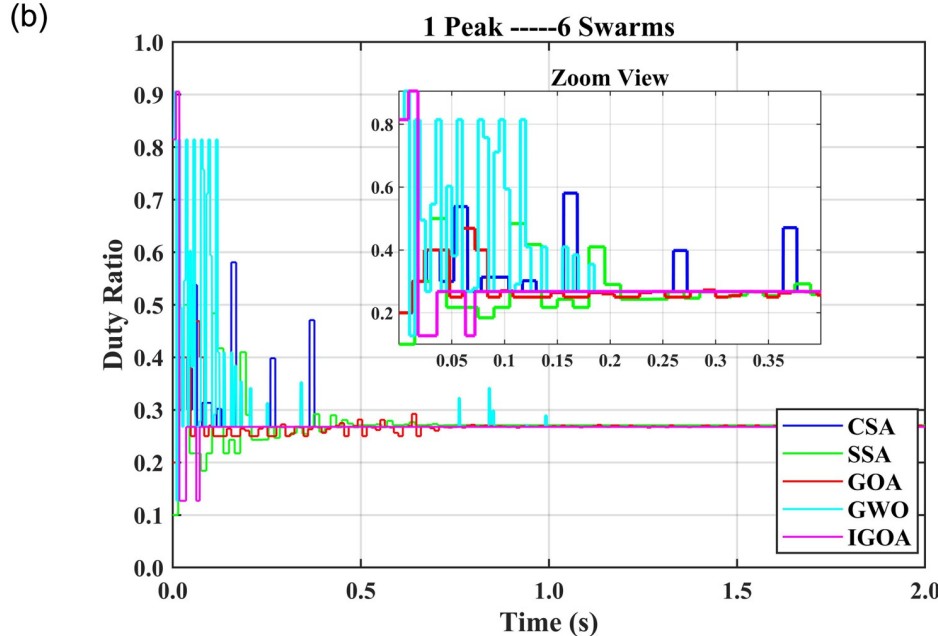

**Fig 9. (a) output powers for 1P6S setup and (b) duty ratios for 1P6S setup.**

average results of the simulations are shown in Table 2. The average FR of all SITs for 4 swarms is greater than 0% except IGOA. IGOA becomes the first approach to achieve 0% FR with four swarms, while other SITs achieve FR varying from 3% to 7%. Furthermore, IGOA significantly reduces tracking and settling times for all the swarms under study compared to all other SITs. It demonstrates the superiority of IGOA over the other SITs in comparison when utilized as an MPPT for PV systems under PSCs. Furthermore, if the configuration of

the PV array ensures that the number of peaks in the P-V curves is at least four, it is preferable to employ four search agents. IGOA's longest tracking duration is less than 0.14 s, which is much less time than the other SIT. It is important to note that the tracking time and failure rate of IGOA are consistent for all peak counts for a particular swarm size; for instance, with six swarms, the tracking time for 4 peaks is 0.15 seconds, and for 5 and 6 peaks it is 0.14 seconds. This indicates that it is not necessary to change the swarm size for varying peaks. The other SITs do not display this functionality. Moreover, it is evident that the tracking and settling time increase with the growing number of swarms and PV array size. However, the tracking accuracy increases with increasing swarms.

The selection of SIT parameters plays an important role in their optimization performance, as shown in Table 3. The values of the optimized parameters of the CSA and the respective equations are taken from [23]. The parameters and equations selected for the SSA employed in this work are carried from [29]. The upper and lower bound of the variables are 0.98 and 0 respectively, and $c2$ is the random number. Similarly, the parameters and equations for GWO are obtained from [27] in which the $a$ linearly decreases from 2 to 0.5. Moreover, $f$ and $l$ for IGOA are calculated from Eqs (7) and (8).

## 5.5 Statistical analysis of SITs

Statistical analysis is also performed using standard deviation (SD), relative error (RE), mean absolute error (MAE), and root mean square error (RMSE). In (11)–(14), RE, MAE, RMSE, and SD are used to measure the sensitivity of the SITs used in this study.

$$E_{RE} = \frac{\sum_{j=1}^{m}(P_{stc} - P_t)}{P_t} \tag{11}$$

$$E_{MAE} = \frac{\sum_{j=1}^{m}(P_{stc} - P_t)}{m} \tag{12}$$

$$E_{RMSE} = \sqrt{\frac{\sum_{j=1}^{m}(P_{stc} - P_t)^2}{m}} \tag{13}$$

$$SD = \sqrt{\frac{\sum_{j=1}^{m}(P_t - P_{mean})^2}{m}} \tag{14}$$

$P_{stc}$ represents the output power at STC, $P_t$ represents the power tracked by the MPPT algorithm, and $m$ represents the total number of simulation runs which is 100 in our case. Statistical data for the average RE, MAE, RMSE, and SD of all cases are graphically represented in Fig 10 for the 5P6S setup. IGOA has the lowest average MAE (average error from the GMPP) among the compared algorithms. The IGOA has a lower average RMSE (how focused the data are on the GMPP) than the other methods, indicating that the power tracked is dense around the GMPP. CSA exhibits the more scattered data around the MPP clearly stated by the RMSE. The SD for case 1 is higher than that in case 2 and case 3, indicating when there is more shading on the PV array, the SD is higher and vice versa. It means that under more shading, the algorithm's output is more likely to depart from the mean value. SD starts to decrease as the irradiance approaches STC, which can be observed in case 3. The IGOA is less affected by a higher value of shading as compared to other algorithms. As a result, its performance is more consistent than that of the other MPPT algorithms used in this study.

**Table 3. Parameters of the algorithms.**

| CSA [23] | SSA [29] | GWO [27] | GOA [33] | Proposed IGOA |
|---|---|---|---|---|
| $\alpha = 0.1$ | $U_b = 0.98$ | $a_u = 2$ | $c_u = 1$ | $c_u = 1$ |
| $\beta = 1.5$ | $L_b = 0$ | $a_l = 0.5$ | $c_l = 0.00004$ | $c_l = 0.00004$ |
| $p_a = 0.25$ | $c2$ = random number | $\epsilon = 4$ | $f = 0.5$ and $l = 1.5$ | $f$ and $l$ from Eqs (7) and (8) |

## 5.6 Comparison with published research work

To increase the breadth of the comparative study, the proposed IGOA is compared with published research works based on SITs for MPPT under PSC. SITs employ random numbers to avoid getting stuck at nearby local peaks. Therefore, the final convergence point is different each time the algorithm is run, even if the initial values are the same. Therefore, statistical

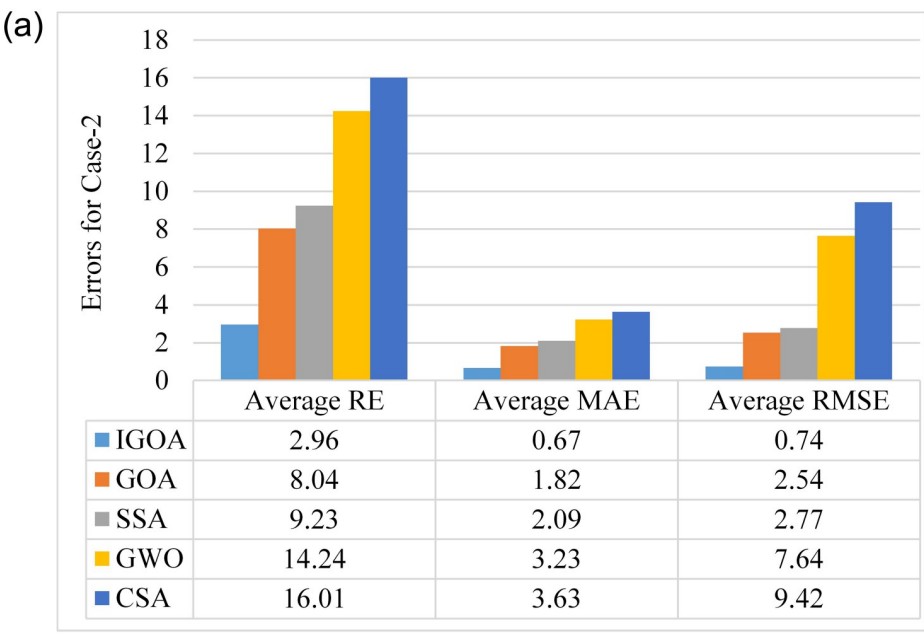

| (a) | Average RE | Average MAE | Average RMSE |
|---|---|---|---|
| IGOA | 2.96 | 0.67 | 0.74 |
| GOA | 8.04 | 1.82 | 2.54 |
| SSA | 9.23 | 2.09 | 2.77 |
| GWO | 14.24 | 3.23 | 7.64 |
| CSA | 16.01 | 3.63 | 9.42 |

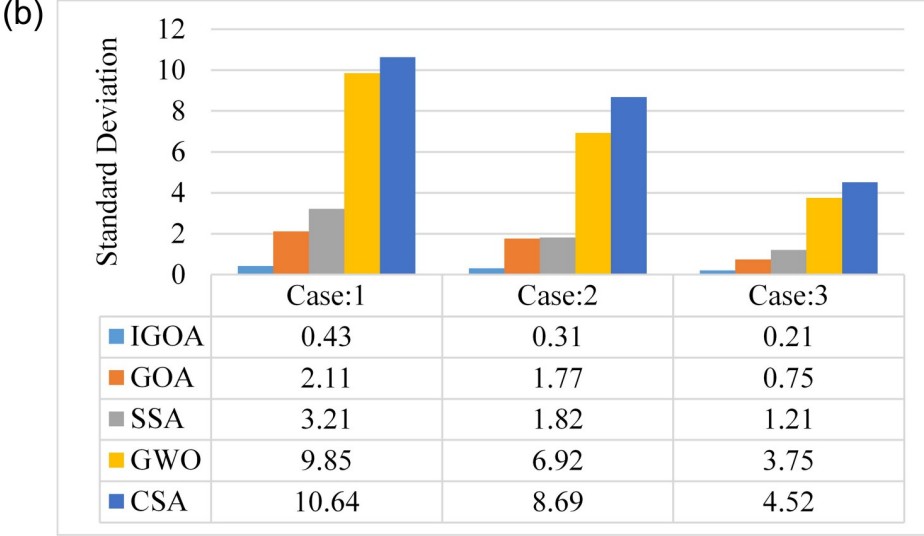

| (b) | Case:1 | Case:2 | Case:3 |
|---|---|---|---|
| IGOA | 0.43 | 0.31 | 0.21 |
| GOA | 2.11 | 1.77 | 0.75 |
| SSA | 3.21 | 1.82 | 1.21 |
| GWO | 9.85 | 6.92 | 3.75 |
| CSA | 10.64 | 8.69 | 4.52 |

**Fig 10. (a) RE, MAE, RMSE for 5P6S setup and (b) standard deviation.**

**Table 4. Comparison of proposed IGOA with published research work.**

| Algorithm / Attribute | Avg. tracking time (s) | Avg. settling time (s) | Avg. efficiency (%) | No. of swarms (No.) |
|---|---|---|---|---|
| GOA-IC [33] | 1.87 | 1.97 | 99.85 | 5 |
| Proposed IGOA | 0.16 | 0.18 | 99.96 | 5 |
| GOA1 [35] | 2.95 | 3.10 | 99.66 | – |
| Proposed IGOA | 0.13 | 0.15 | 99.96 | 4 |
| GOA2 [30] | case 2: 0.32; case 3: 0.16 | case 2: 0.52; case 3: 0.32 | 99.92 | 4 |
| Proposed IGOA | 0.13;0.12 | 0.18;0.19 | 99.97 | 4 |
| ICS [23] | 0.28 | – | 99.97 | 4 |
| Proposed IGOA | 0.11 | 0.13 | 99.97 | 4 |
| IGWO [27] | 0.35 | – | 99.81 | – |
| Proposed IGOA | 0.13 | 0.14 | 99.97 | 4 |
| IBA [26] | 3.60 | – | 99.85 | 5 |
| Proposed IGOA | 0.16 | 0.17 | 99.96 | 5 |
| ISSAPSO [29] | 0.37 | – | 99.57 | – |
| Proposed IGOA | 0.21 | 0.22 | 99.95 | 4 |

results are obtained by running the simulation 100 times and taking the average as given in Table 4. However, the results shown in the figures are for a single MPPT run, not an average of 100 runs.

A comparison of average tracking time, settling time, and average efficiency is presented between the proposed IGOA and the previously published research work such as improved salp swarm algorithm based on PSO (ISSAPSO) [29], the GOA for PSC [35], GOA for complex PSC [30], improved bat algorithm (IBA) [26], improved grey wolf optimization (IGWO) [27], improved cuckoo search algorithm (ICS) [23], and GOA with incremental conductance (GOA-IC) [33]. For a fair comparison, the same environmental conditions, such as irradiance, the solar photovoltaic panels used, and the same number of swarms (particles), were considered. IGOA is compared with [26, 33] using 5 particles as mentioned in their work and employing 4 particles when compared with [23, 30].

Fig 11a compares the power of proposed IGOA with ISSAPSO [29]. In the simulation, four series solar modules from the Tata Power Solar System TP250MBZ (250 W) are taken using the irradiance level [500 1000 800 1000] at 25°C, whose global peak (GP) is 640 W. The proposed technique tracks the GP in 0.21 s as compared to 0.37 s in ISSPSO, which means the proposed method is 43% faster than its counterpart. In addition, an oscillation-free power curve is obtained for IGOA, which confirms its superiority.

Fig 11b depicts power graphs for the proposed IGOA and GOA [35]. Six CENTSYS solar modules (40 W) are used in a series-parallel combination using the irradiance level [800 100 1000 500 400 1000] at 25°C having GP at 140.2 W. As can be observed, the proposed IGOA tracks the GP in 0.13 s as compared to 2.95 s for GOA which confirms the outperformance of the proposed method as it follows the GP 14 times quicker.

The proposed IGOA is compared to GOA [30]. A case (case:2) is taken for comparison in which four SunPower SPR-315E-WHT-D (315 W) PV modules are connected in series with an irradiance level of [800 250 700 400] at 25°C and the GP is 450 W. The power graphs for both analogies are given in Fig 11c which demonstrates that the settling time for IGOA is 0.18 s as compared to 0.52 s for GOA. This means that IGOA is 65% faster than GOA. Moreover, an oscillation-free power curve is obtained for the IGOA, which confirms its supremacy.

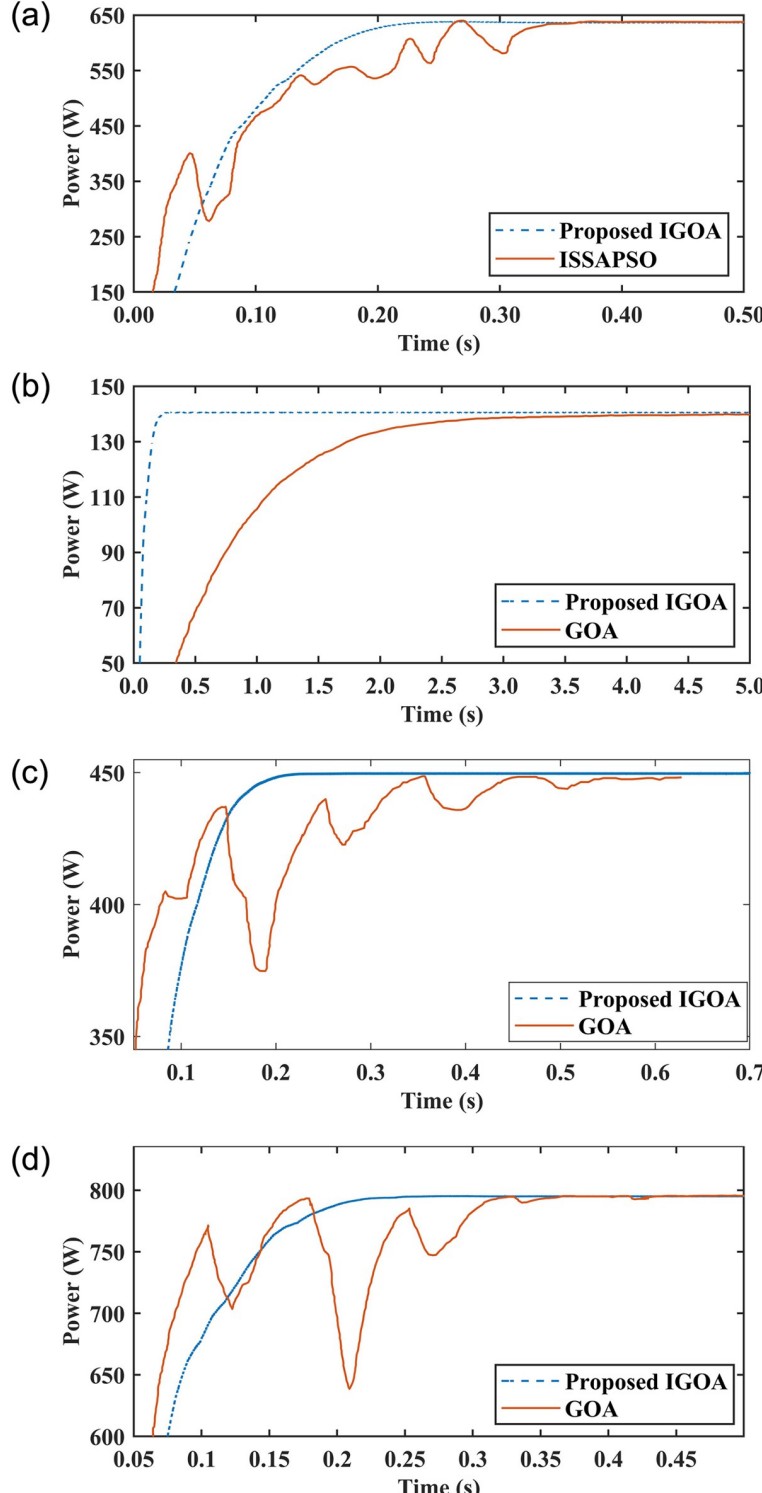

**Fig 11.** (a) Comparison with ISSAPSO [29], (b) GOA [35], (c) case 2 of [30] and (d) case 3 of [30].

IGOA is compared with another case (case:3) of GOA [30]. GP is 796 W when four PV modules are combined in series at an irradiance level of [500 800 1000 900] at 25˚C. Fig 11d clearly shows that the settling time for IGOA is 0.18 s as compared to 0.32 s for GOA, indicating that the proposed approach is 43% faster and more effective than GOA. In addition, an oscillation-free power curve for IGOA is obtained, demonstrating its superiority over GOA.

There are considerable disparities in the tracking times among the compared SITs. It can be seen in Table 4 that the tracking time is 1.87 s [33], 0.28 s [23], 0.35 s [27], 3.6 s [26], and the proposed IGOA tracks the GP in 0.16 s, 0.11 s, 0.13 s and 0.16 s respectively. A comparative analysis of various numbers of swarms ranging from 3 to 10 shows that the tracking time varies between 0.18 s and 0.74 s [23]. The proposed IGOA tracks GMPP about 15 times faster than [33], and 30 times faster than [26]. It takes the shortest time to reach GP, followed by ICS [23], IGWO [27], GOA-IC [33], and IBA [26].

It can be seen in Table 4 that the settling time of the proposed work is less than the other techniques like GOA-IC [33], GOA1 [35] and GOA2 [30]. The efficiency obtained by the proposed work has been improved compared to the existing work, except for the one given against reference [23]. However, it should be mentioned that the same efficiency was obtained with the proposed work in less time, that is, 0.11 s as compared to 0.28 s for [23].

## 6 Conclusion

This work is mainly focused on reducing the tracking time and start-up oscillations under PSC while tracking an MPP in a PV system. In the previous work using SITs, the tracking time and oscillations were high under PSC. The improved tracking ability of the proposed IGOA-based MPPT scheme has been validated by comparing its performance with well-known SITs such as GOA, ISSAPSO, SSA, GWO, and CSA with the help of four different irradiance profiles and swarm sizes. Comparative results and statistical analysis show a clear advantage of IGOA over its counterparts that exhibit faster tracking speed, shorter settling time, reduced startup oscillations, and lower FR. IGOA guarantees a tracking time of 0.07-0.15 s, which is 2 to 24 times faster than other SITs, allowing quick adaptability to changes in atmospheric conditions. Unlike other SITs, it shows a settling time of 0.08-0.17 s which is 6 to 13 times shorter, and shows that varying the number of peaks due to shading conditions has no impact on settling time for a specific swarm size. It ensures high reliability in MPP tracking with less than 2% FR in the 3 peaks system for all swarms and 0% FR for the rest of the setup. This unique characteristic highlights the robustness of the IGOA and distinguishes it from other SITs. Increasing the swarm agents leads to a reduction in FR of IGOA, which highlights its scalability and ensures a higher possibility of finding acceptable solutions.

## Author Contributions

**Conceptualization:** Muhammad Shahid Wasim, Muhammad Amjad.

**Data curation:** Muhammad Shahid Wasim.

**Formal analysis:** Muhammad Shahid Wasim, Muhammad Amjad.

**Investigation:** Muhammad Shahid Wasim, Muhammad Amjad.

**Methodology:** Muhammad Shahid Wasim, Muhammad Amjad, Muhammad Abbas Abbasi.

**Project administration:** Muhammad Amjad.

**Resources:** Muhammad Shahid Wasim, Muhammad Amjad, Muhammad Abbas Abbasi.

**Software:** Muhammad Shahid Wasim, Muhammad Amjad.

**Supervision:** Muhammad Amjad, Muhammad Abbas Abbasi, Abdul Rauf Bhatti, Akhtar Rasool.

**Validation:** Muhammad Shahid Wasim, Muhammad Amjad, Muhammad Abbas Abbasi, Abdul Rauf Bhatti, Akhtar Rasool.

**Visualization:** Muhammad Amjad, Abdul Rauf Bhatti, Akhtar Rasool.

**Writing – original draft:** Muhammad Shahid Wasim, Muhammad Amjad, Muhammad Abbas Abbasi.

**Writing – review & editing:** Muhammad Amjad, Abdul Rauf Bhatti, Akhtar Rasool.

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
