## [Decision Letter · Decision Letter 0]

30 Jun 2023

PONE-D-23-11122An Improved Grasshopper-based MPPT Approach to Reduce Tracking Time and Startup Oscillations in Photovoltaic System under Partial Shading ConditionsPLOS ONE

Dear Dr. RASOOL,

Thank you for submitting your manuscript to PLOS ONE. After careful consideration, we feel that it has merit but does not fully meet PLOS ONE’s publication criteria as it currently stands. Therefore, we invite you to submit a revised version of the manuscript that addresses the points raised during the review process.

We look forward to receiving your revised manuscript.

Kind regards,

Ali Wagdy Mohamed

Academic Editor

PLOS ONE

Journal Requirements:

Additional Editor Comments (if provided):

The authors must consider the following recent papers. Thus, they must compare with the most recent algorithms and follow the same presentation style, experimentation and analsys.

1) Optimal identification of unknown parameters of photovoltaic models using dual-population gaining-sharing knowledge-based algorithm.

2)A new method for parameter extraction of solar photovoltaic models using gaining–sharing knowledge based algorithm.

Reviewers' comments:

Reviewer's Responses to Questions

**Comments to the Author**

1. Is the manuscript technically sound, and do the data support the conclusions?

Reviewer #1: Partly

Reviewer #2: Yes

2. Has the statistical analysis been performed appropriately and rigorously? 

Reviewer #1: Yes

Reviewer #2: No

3. Have the authors made all data underlying the findings in their manuscript fully available?

Reviewer #1: Yes

Reviewer #2: No

4. Is the manuscript presented in an intelligible fashion and written in standard English?

Reviewer #1: Yes

Reviewer #2: Yes

5. Review Comments to the Author

Reviewer #1: The authors presented work titled "An Improved Grasshopper-based MPPT Approach to Reduce Tracking Time and Startup Oscillations in Photovoltaic System under Partial Shading Conditions". There are some improvements needs to be done before final publishing. The comments are:

1- Contributions and Paper Organization section should be presented at the end of introduction.

2- The simulation setup should be elaborated in detail.

3- More up-to-date references should be added in the literature.

4- The results and discussion section needs to elaborated more.

5- The comparative analysis with the other latest presented techniques should be presented.

Reviewer #2: Title

PONE-D-23-11122

An Improved Grasshopper-based MPPT Approach to Reduce Tracking Time and Startup Oscillations in Photovoltaic System under Partial Shading Conditions

Summary:

The article proposed an improved grasshopper optimization algorithm and its application for MPPT control under partial shading condition. The results in simulation setup show incremental improvement to the existing literature under different case studies.

Comments:

The results draw only rely on the software simulations and lack experimental study. The improvements to the existing literature should be highlighted in terms of mathematical modeling and test function results. The objective function is said to be the duty ratio in Eq. (9) needs further explanation. Line 218 take as the target position. However how the target is actively updated in not clear. The manuscript overall lacks the clear innovation. Needs major improvements. There are some typos and writeup errors. i.e. Where� where, etc.

6. PLOS authors have the option to publish the peer review history of their article (what does this mean?). If published, this will include your full peer review and any attached files.

Reviewer #1: No

Reviewer #2: No

---

## [Author Response · Author response to Decision Letter 0]

11 Jul 2023

Editor Concerns:

The authors must consider the following recent papers. Thus, they must compare with the most recent algorithms and follow the same presentation style, experimentation, and analyses.

1) Optimal identification of unknown parameters of photovoltaic models using dual-population gaining-sharing knowledge-based algorithm.

2) A new method for parameter extraction of solar photovoltaic models using gaining–sharing knowledge-based algorithm.

Author Response: Thank you for your valuable feedback. We appreciate your suggestion to consider recent papers while following a similar presentation style, experimentation, and analysis. 

We have taken this suggestion into account and have incorporated references [47, 48] to the suggested papers in our revised manuscript. By incorporating these references, we aim to provide a comprehensive overview of the state-of-the-art algorithms in our field and ensure that our work is aligned with the latest advancements.

47. Xiong G, Li L, Mohamed AW, Zhang J, Zhang Y, Chen H, et al. Optimal identification of unknown parameters of photovoltaic models using dual-population gaining-sharing knowledge-based algorithm. International Journal of Intelligent Systems. 2023;2023.

48. Xiong G, Li L, Mohamed AW, Yuan X, Zhang J. A new method for parameter extraction of solar photovoltaic models using gaining–sharing knowledge-based algorithm. Energy Reports. 2021; 7:3286–3301.

Once again, we thank you for your insightful comments, which have greatly contributed to improving the quality of our manuscript.

Reviewer #1 Concerns: 

The authors presented a work titled "An Improved Grasshopper-based MPPT Approach to Reduce Tracking Time and Startup Oscillations in Photovoltaic System under Partial Shading Conditions". There are some improvements that need to be made before final publishing. The comments are:

Reviewer#1, Concern # 1: Contributions and Paper Organization section should be presented at the end of introduction.

Author Response: Authors are thankful to the respected reviewer for putting the effort into carefully reading the manuscript and giving valuable suggestions to improve the level of the manuscript. 

To address the concern, we have added contributions and paper organization section at the end of the introduction in revised manuscript and can be found on page 3, line 115-129 red highlighted text. 

Comparative results and statistical analysis show a clear advantage of IGOA over its counterparts that exhibit faster tracking speed, shorter settling time, reduced startup oscillations, and lower FR. IGOA guarantees a tracking time of 0.07-0.15 second, which is 2 to 24 times faster than other SITs, allowing quick adaptability to change in atmospheric conditions. Unlike other SITs, it shows a settling time of 0.08-0.17 second which is 6 to 13 times shorter and shows that varying the number of peaks due to shading conditions has no impact on settling time for a specific swarm size. It ensures high reliability in MPP tracking with less than 2% FR in the 3 peaks system for all swarms and 0% FR for the rest of the setup. This unique characteristic highlights the robustness of the IGOA and distinguishes it from other SITs. Increasing the swarm agents leads to a reduction in FR of IGOA, which highlights its scalability and ensures a higher possibility of finding acceptable solutions. The remaining paper is arranged as follows. The effect of PSCs on the PV system is described in Section 2. Section 3 explores the proposed method. The simulation setup is presented in Section 4. Section 5 presents the results and Section 6 is reserved for the conclusion.

Reviewer#1, Concern # 2: The simulation setup should be elaborated in detail. 

Author Response: Thank you very much for raising the concern. It is our mistake that we could not provide the details on this point in our previously submitted manuscript. 

However, by considering the concern of respected reviewer, we have added and elaborated another section (Section 4) named “Simulation setup”. The detail is provided with the help of a new figure (Fig 5) in the revised manuscript on page 8, line 238-252 red highlighted.

4 Simulation Setup

A simple model is taken to evaluate the viability of algorithms under various PSCs, as shown in Fig 5. The PV array receives different levels of irradiance (G) from the visible light spectrum, which results in current and, ultimately, electricity [47,48]. The generated power enters a boost converter, which modifies the output voltage according to the DC link voltage level. A tiny capacitor (C) is inserted in between the PV array and the boost converter to supply the ripple current required for opening and closing the converter's MOSFET. SITs are used one by one to monitor PV current and voltage to track the system's MPP. Sun-Power PV modules (SPR-315E-WHT-D) are connected in series to make an array. The performance of the system is evaluated for one unshaded condition and three PSCs provided in Table 1 that have 4 to 6 peaks in their characteristic curves shown in Fig 2. All peaks are evaluated for various numbers of swarms ranging from 3 to 6. The simulation is run 100 times to combat the unpredictable behavior of SITs.

Reviewer#1, Concern # 3: More up-to-date references should be added to the literature. 

Author Response: Thank you for your valuable suggestion. We have included two up-to-date references [47, 48] in the revised manuscript.

47. Xiong G, Li L, Mohamed AW, Zhang J, Zhang Y, Chen H, et al. Optimal identification of unknown parameters of photovoltaic models using dual-population gaining-sharing knowledge-based algorithm. International Journal of Intelligent Systems. 2023;2023.

48. Xiong G, Li L, Mohamed AW, Yuan X, Zhang J. A new method for parameter extraction of solar photovoltaic models using gaining–sharing knowledge-based algorithm. Energy Reports. 2021; 7:3286–3301.

Reviewer#1, Concern # 4: The results and discussion section need to elaborate more.

Author Response: Thank you very much for raising the concern. It is our mistake that we could not provide the detail on this point in our previously submitted manuscript. 

However, by considering the concern of respected reviewer, we have added a separate section (Section 5) named “Results and discussion” and elaborated the concern in the revised manuscript on page 9, line 253-267, red highlighted. To add more details, Section 5 is subdivided into five new subsections named as,

5 Results and discussion

The proposed IGOA is compared with four other SITs: CSA, GOA, GWO and SSA in terms of tracking and settling times, failure rate (FR) and startup oscillations. The FR is stated as the ratio between the number of attempts maturing at local peaks to the total number of attempts during the simulations. Some of the algorithms may perform well for fewer peaks (fewer shading conditions), and others may perform well under more shading conditions. Similarly, the number of swarms may also affect the performance of the system. Therefore, for a fair comparison, these techniques are studied for 1, 4, 5, and 6 peaks with swarm sizes of 3, 4, 5, and 6. Table 2 depicts the comparison results. The following subsections present the comparative performance analysis of SITs for various peaks and swarm numbers.

5.1: Comparison of SITs for four peaks with multiple agents 

In this section, the comparison among SITs is performed for four peaks and 3, 4, 5, and 6 swarm agents.

(Improvement for this subsection is added on page #10, line 279-286, blue highlighted text)

When the swarm setting was changed to 4P6S, all SITs maintained 0% FR, but with an increase in oscillations except IGOA. Fig 6a shows the fitness value (power obtained) over time and Fig 6b depicts the graph between time and duty ratios for the 4P6S setup. It is evident that the duty rate for IGOA exhibits a rapid increase and tracks the GMPP within 0.078 second and settles the output in 0.15 second. This indicates that IGOA achieves fitness values faster than the other algorithms as time progresses. On the other hand, other algorithms show more oscillations and more time for convergence. Compared to IGOA, their duty ratios take longer to find the best solution.

5.2: Comparison of SITs for five peaks with multiple agents

(Improvement for this subsection is added on page #10-11, line 306-313, blue highlighted text)

For six swarms (5P6S) all the SITs show 0% FR except CSA and GWO which still have 1% FR. Fig 6a shows the fitness value (power obtained) over time and Fig6b depicts the graph between time and duty ratios for the 5P6S setup. It is evident that the duty rate for IGOA exhibits a rapid increase and tracks the GMPP within 0.07 second and settles the output in 0.14 second. On the other hand, tracking time for SSA is 1.49 second, 1.37 second for GWO, 1.39 second for CSA, and 0.39 second for GOA and show more oscillations. This indicates that IGOA achieves fitness values faster than the other algorithms as time progresses. Compared to IGOA, their duty ratios take longer to find the best solution.

5.3: Comparison of SITs for six peaks with multiple agents

(Improvement for this subsection is added on page #11-12, line 329-335, blue highlighted text)

For 6P6S SSA, GOA and IGOA maintained 0% FR. Fig 8a shows the fitness value (power obtained) over time and Fig 8b depicts the graph between time and duty ratios for the 6P6S setup. It is evident that the duty rate for IGOA exhibits a rapid increase and tracks the GMPP within 0.07 second and settles the output in 0.16 second. On the other hand, settling time for CSA is 1.66 second, GWO 1.85 second, SSA 1.89 second, and 1.63 second for GOA. This indicates that IGOA achieves fitness values faster than the other algorithms as time progresses. Compared to IGOA, their duty ratios take longer to find the best solution.

5.4: Comparison of SITs for single peak with multiple agents

(Improvement for this subsection is added on page #12, line 351-359, blue highlighted text)

When the swarm setting changed to 1P6S, all SITs maintained 0% FR. Fig 9a shows the fitness value (power obtained) over time and Fig 9b depicts the graph between time and duty ratios for the 1P6S setup. It is evident that the duty rate for IGOA exhibits a rapid increase and tracks the GMPP within 0.03 second and settles the output in 0.07 second. This indicates that IGOA achieves fitness values faster than the other algorithms as time progresses. On the other hand, other algorithms show more oscillations and more time for convergence. Compared to IGOA, their duty ratios take longer to find the best solution.

5.5: Statistical analysis of SITs

Reviewer#1, Concern # 5: The comparative analysis with the other latest presented techniques should be presented.

Author Response: Authors are thankful to the respected reviewer for putting the effort into carefully reading the manuscript and giving valuable suggestions to improve the level of the manuscript. 

It is our mistake that we could not provide the detail on this point in our initial submission. We would like to clarify that we have indeed included a thorough comparison in terms of tracking time, settling time, startup oscillations and failure rate with the most recent published swarm intelligence optimization technique considering MPPT in solar system such as salp swarm algorithm (SSA) in addition to grasshopper optimization algorithm (GOA) in our manuscript.

The comparison with SSA is highlighted in Table 2 on page 13 and with ISSAPSO in Table 4 on page 16 considering MPPT published in year 2022. 

Reviewer#2 Concerns: 

The article proposed an improved grasshopper optimization algorithm and its application for MPPT control under partial shading condition. The results in simulation setup show incremental improvement to the existing literature under different case studies.

Reviewer#2, Concern # 1: The results draw only rely on the software simulations and lack experimental study. The improvements to the existing literature should be highlighted in terms of mathematical modeling and test function results.

Author Response: Authors are thankful to the respected reviewer for putting the effort into carefully reading the manuscript and giving valuable suggestions to improve the level of the manuscript. 

We have incorporated the suggested improvements into our revised manuscript. We have included and explained an extra equation (Equation 9) in addition to other equations 6-8, to provide improvements to the existing literature in terms of mathematical modeling. The explanation can be found on page # 7 line 195-208 in red color and copied below too.

Therefore, it is necessary to redefine the range of f and l in the distance range [0,1] as the maximum distance between the two GH (duty ratio) is 1. Therefore, social interaction (‘s’ function) is modified to (6) to balance the two forces. 

s(r)= f_new exp^((-r)/l_new ) -exp^(-r) (6)

Here f_new and l_new are defined in (7) and (8) respectively, while f and l retain their previous values, that is, 0.5 and 1.5, respectively. Moreover, for intermittent environmental conditions a normalized average irradiance factor G_N is added in the equations to get f_new in between [0.65,0.9] and l_new in the range [2,2.9].

f_new=f(G_N+x) (7)

l_new=l(G_N+x) (8)

Where x is from one of the following two cases.

Case 1: if (0<G_N≤0.5) then x=1.3

Case 2: if (0.5< G_N≤1) then x=0.8

The position equation becomes (9) in which d_i is the position of i_th GH and d_u and d_l represent its upper and lower limits respectively.

d_i= c∑_(j=1,j≠i)^N▒c ((d_u-d_l)/2 s(‖d_j-d_i ‖) (d_j-d_i)/D_ij ) +T ^_d (9)

Moreover, we have conducted extensive analysis regarding the test function results. The revised manuscript now includes a detailed discussion highlighted on: 

page 10, line 279-286, blue highlighted text,

When the swarm setting was changed to 4P6S, all SITs maintained 0% FR, but with an increase in oscillations except IGOA. Fig 6a shows the fitness value (power obtained) over time and Fig 6b depicts the graph between time and duty ratios for the 4P6S setup. It is evident that the duty rate for IGOA exhibits a rapid increase and tracks the GMPP within 0.078 second and settles the output in 0.15 second. This indicates that IGOA achieves fitness values faster than the other algorithms as time progresses. On the other hand, other algorithms show more oscillations and more time for convergence. Compared to IGOA, their duty ratios take longer to find the best solution.

page 10-11, line 306-313, blue highlighted text,

For six swarms (5P6S) all the SITs show 0% FR except CSA and GWO which still have 1% FR. Fig 7a shows the fitness value (power obtained) over time and Fig 7b depicts the graph between time and duty ratios for the 5P6S setup. It is evident that the duty rate for IGOA exhibits a rapid increase and tracks the GMPP within 0.07 second and settles the output in 0.14 second. On the other hand, tracking time for SSA is 1.49 second, 1.37 second for GWO, 1.39 second for CSA, and 0.39 second for GOA and show more oscillations. This indicates that IGOA achieves fitness values faster than the other algorithms as time progresses. Compared to IGOA, their duty ratios take longer to find the best solution.

page 11-12, line 329-335, blue highlighted text,

For 6P6S SSA, GOA and IGOA maintained 0% FR. Fig 8a shows the fitness value (power obtained) over time and Fig 8b depicts the graph between time and duty ratios for the 6P6S setup. It is evident that the duty rate for IGOA exhibits a rapid increase and tracks the GMPP within 0.07 second and settles the output in 0.16 second. On the other hand, settling time for CSA is 1.66 second, GWO 1.85 second, SSA 1.89 second, and 1.63 second for GOA. This indicates that IGOA achieves fitness values faster than the other algorithms as time progresses. Compared to IGOA, their duty ratios take longer to find the best solution.

page 12, line 351-359, blue highlighted text.

When the swarm setting changed to 1P6S, all SITs maintained 0% FR. Fig 9a shows the fitness value (power obtained) over time and Fig 9b depicts the graph between time and duty ratios for the 1P6S setup. It is evident that the duty rate for IGOA exhibits a rapid increase and tracks the GMPP within 0.03 second and settles the output in 0.07 second. This indicates that IGOA achieves fitness values faster than the other algorithms as time progresses. On the other hand, other algorithms show more oscillations and more time for convergence. Compared to IGOA, their duty ratios take longer to find the best solution.

We believe that these revisions significantly show the improvements with respect to test function results. Thank you for your guidance, which has greatly improved the quality of our work. 

Reviewer#2, Concern # 2: The objective function is said to be the duty ratio in Eq. (9) needs further explanation. 

Author Response: Thank you for your valuable feedback on our manuscript. We appreciate your input and have made the necessary revisions to address the concerns raised.

We apologize for the lack of clarity in our previous explanation regarding the objective function stated as the duty ratio in Eq. (9), now the Eq (10). We have provided further explanation and clarification regarding the duty ratio d as the objective function on page # 7, line 212-222. The same is reproduced here…

Here, di is the position of ith GH that is to be optimized. The algorithm aims to find the optimal d that maximizes the power received by the system. At each iteration, the power received for a specific di(l) is evaluated. This power value represents the fitness or objective value of the GH. The algorithm compares this power value in the current iteration l with the power value in the previous iteration l-1. The inequality in (10) serves as a criterion to determine whether the GHs have improved their positions d in terms of maximizing the power received. If the power received in the current iteration is greater than the power received in the previous iteration, this implies that the GHs have made progress and have moved towards a more optimal d. Repetition of this process over several iterations is key to moving GHs towards the optimal d and thus reaching the maximum output power target.

We believe that these revisions significantly improve the clarity and understanding of our work, providing a more comprehensive explanation of the objective function.

Reviewer#2, Concern # 3: Line 218 takes the target position. However how the target is actively updated in not clear.

Author Response: Thank you for your valuable feedback on our manuscript. We appreciate your input and have made the necessary revisions to address the concerns raised.

It is our mistake that we could not provide clarity in our previous explanation regarding active update of the target position. We have addressed your concern in our revised manuscript by providing a detailed description of how the target position is actively updated. 

Moreover, we have revised and elaborated flow chart (Fig 4) of the proposed algorithm to add more clarity on updating the target position and can be found on page 8 line 223-236. The same is reproduce here…

Flow chart of the IGOA is shown in Fig 4. Initially, the IGOA initializes all its parameters and randomly generates the initial positions (d¬i) of each GH within the limits. After that, it computes the fitness value (output power) of each GH based on the objective function (10). Then, it identifies and stores the best fitness value (highest output power) and the corresponding d as the target position. After this process, the iterative cycle starts and updates the variables with their respective equations as depicted in the flow chart. Then it recalculates the fitness values for the updated positions and compares this with the best fitness value obtained so far. If any GH has a better fitness value, it updates the best fitness value accordingly. It completes its iterative cycle using the equations specified in the flow chart. When the current iteration reaches its maximum limit, it checks the change in irradiance. If there is a change, it repeats the whole process otherwise it terminates. After termination, it returns the best target position which represents the optimized d that satisfies the objective function (10) with maximum output power.

We believe that these revisions significantly improve the clarity and understanding of our work, providing a more comprehensive explanation on active update of the target position.

Reviewer#2, Concern # 4: The manuscript overall lacks clear innovation. Needs major improvements. 

Author Response: Thank you for your valuable feedback on our manuscript. We appreciate your input and have made the necessary revisions to address the concerns raised.

It is our mistake that we could not provide clarity in our previous explanation regarding active update of the target position. We have addressed your concern in our revised manuscript by providing clarity on the research work and can be found on page 16, line 469-480 red highlighted text.

Comparative results and statistical analysis show a clear advantage of IGOA over its counterparts that exhibit faster tracking speed, shorter settling time, reduced startup oscillations, and lower FR. IGOA guarantees a tracking time of 0.07-0.15 second, which is 2 to 24 times faster than other SITs, allowing quick adaptability to change in atmospheric conditions. Unlike other SITs, it shows a settling time of 0.08-0.17 second which is 6 to 13 times shorter and shows that varying the number of peaks due to shading conditions has no impact on settling time for a specific swarm size. It ensures high reliability in MPP tracking with less than 2 % FR in the 3 peaks system for all swarms and 0% FR for the rest of the setup. This unique characteristic highlights the robustness of the IGOA and distinguishes it from other SITs. Increasing the swarm agents leads to a reduction in FR of IGOA, which highlights its scalability and ensures a higher possibility of finding acceptable solutions.

Moreover, in our manuscript, we have devoted considerable effort to address several crucial aspects, including tracking time, settling time, failure rate, and startup oscillations during MPPT process using IGOA. These aspects are critical for evaluating the effectiveness of optimization algorithms. 

Firstly, we have significantly reduced the tracking time. IGOA now exhibits faster convergence and more efficient exploration of the search space.

Secondly, the settling time has been notably improved. Our work has minimized fluctuations commonly observed during the optimization process, allowing IGOA to converge more steadily and reach a stable solution in a shorter timeframe.

Furthermore, we have effectively mitigated startup oscillations, which refer to the initial instability and erratic behavior exhibited by optimization algorithms during the initial iterations. By minimizing these oscillations, we have ensured a smoother and more consistent optimization process from the beginning, leading to improved overall performance.

Lastly, we have addressed the issue of failure rate, which indicates the likelihood of an algorithm failing to find an acceptable solution within a specified number of iterations. We have significantly reduced the failure rate, thereby increasing its reliability and applicability in various optimization scenarios.

Overall, our IGOA represents a notable contribution to the field of MPPT. Through rigorous comparative analysis, we have demonstrated that IGOA outperforms other popular algorithms such as CSA, SSA, and GWO in terms of tracking time, settling time, failure rate, and startup oscillations.

These improvements highlight the innovative nature of our research, as we have successfully addressed key limitations in existing algorithms, leading to enhanced effectiveness in various performance metrics.

Reviewer#2, Concern # 5: There are some typos and writeup errors. i.e., Where� where, etc.

Author Response: Thank you for bringing the typos and writeup errors to our attention. We have thoroughly reviewed and corrected these issues in the revised manuscript to ensure the accuracy and clarity of our findings.

---

## [Decision Letter · Decision Letter 1]

13 Aug 2023

An Improved Grasshopper-based MPPT Approach to Reduce Tracking Time and Startup Oscillations in Photovoltaic System under Partial Shading Conditions

PONE-D-23-11122R1

Dear Dr. RASOOL,

We’re pleased to inform you that your manuscript has been judged scientifically suitable for publication and will be formally accepted for publication once it meets all outstanding technical requirements.

Kind regards,

Ali Wagdy Mohamed

Academic Editor

PLOS ONE

Additional Editor Comments (optional):

Reviewers' comments:

Reviewer's Responses to Questions

**Comments to the Author**

1. If the authors have adequately addressed your comments raised in a previous round of review and you feel that this manuscript is now acceptable for publication, you may indicate that here to bypass the “Comments to the Author” section, enter your conflict of interest statement in the “Confidential to Editor” section, and submit your "Accept" recommendation.

Reviewer #1: All comments have been addressed

2. Is the manuscript technically sound, and do the data support the conclusions?

Reviewer #1: Yes

3. Has the statistical analysis been performed appropriately and rigorously? 

Reviewer #1: Yes

4. Have the authors made all data underlying the findings in their manuscript fully available?

Reviewer #1: Yes

5. Is the manuscript presented in an intelligible fashion and written in standard English?

Reviewer #1: Yes

6. Review Comments to the Author

Reviewer #1: The authors have successfully addressed all comments and the manuscript is now ready to be published.

7. PLOS authors have the option to publish the peer review history of their article (what does this mean?). If published, this will include your full peer review and any attached files.

Reviewer #1: No

---

## [Editor Report · Acceptance letter]

17 Aug 2023

PONE-D-23-11122R1 

An Improved Grasshopper-based MPPT Approach to Reduce 2 Tracking Time and Startup Oscillations in Photovoltaic 3 System under Partial Shading Conditions 

Dear Dr. Rasool:

I'm pleased to inform you that your manuscript has been deemed suitable for publication in PLOS ONE. Congratulations! Your manuscript is now with our production department. 

Kind regards, 

on behalf of

Dr. Ali Wagdy Mohamed 

Academic Editor

PLOS ONE